# In-cloud measurements highlight the role of aerosol hygroscopicity in cloud droplet formation

Olli Väisänen[1], Antti Ruuskanen[2], Arttu Ylisirniö[1], Pasi Miettinen[1], Harri Portin[3], Liqing Hao[1], Ari Leskinen[1,2], Mika Komppula[2], Sami Romakkaniemi[2], Kari E. J. Lehtinen[1,2], Annele Virtanen[1]

[1]University of Eastern Finland, Department of Applied Physics, P.O. Box 1627, 70211 Kuopio, Finland
[2]Finnish Meteorological Institute, P.O. Box 1627, 70211 Kuopio, Finland
[3]Helsinki Region Environmental Services Authority, P.O. Box 100, 00066 HSY, Finland

*Correspondence to*: Annele Virtanen (annele.virtanen@uef.fi)

**Abstract.** The relationship between aerosol hygroscopicity and cloud droplet activation was studied at the Puijo measurement station in Kuopio, Finland, during the autumn 2014. The hygroscopic growth of 80, 120 and 150 nm particles was measured at 90% relative humidity with a hygroscopic tandem differential mobility analyzer. Typically, the growth factor (GF) distributions appeared bimodal with clearly distinguishable peaks around 1.0–1.1 and 1.4–1.6. However, the relative contribution of the two modes appeared highly variable reflecting the probable presence of fresh anthropogenic particle emissions. The hygroscopicity-dependent activation properties were estimated in a case study comprising four separate cloud events with varying characteristics. At 120 and 150 nm, the activation efficiencies within the low- and high-GF modes varied between 0%–34% and 57%–83%, respectively, indicating that the less hygroscopic particles remained mostly non-activated, whereas the more hygroscopic mode was predominantly scavenged into cloud droplets. By modifying the measured GF distributions, it was estimated how the cloud droplet concentrations would change if all the particles belonged to the more hygroscopic group. According to $\kappa$-Köhler simulations, the cloud droplet concentrations increased up to 70% when the possible feedback effects on effective peak supersaturation (between 0.16% and 0.29%) were assumed negligible. This is an indirect but clear illustration of the sensitivity of cloud formation to aerosol chemical composition.

## 1 Introduction

Atmospheric aerosols play a key role in the global climate system. They can affect the Earth's radiation balance either directly by scattering and absorbing the solar radiation, or indirectly, via clouds. Despite their relatively long-known influence pathways, atmospheric aerosols and especially their interactions with clouds, are still recognized as the most important source of uncertainty in the estimates of radiative forcing over the industrial period (IPCC, 2013). Although the climatic sensitivity and clouds are linked via numerous feedback mechanisms, one fundamental key towards a diminished uncertainty is to understand the factors controlling the particle's ability to act as cloud condensation nuclei (CCN).

The particles' ability to activate into cloud droplets at a certain level of supersaturation depends on their size and chemical composition. The role of particle size is already quite well identified and it's commonly considered as the most important factor. For example, according to the early model calculations by Junge and McLaren (1971) and measurements by Fitzgerald (1973), the shape of the CCN spectrum was predominantly determined by the initial particle size distribution, and

compositional variations became significant only when the aerosol was highly insoluble. More recently, Dusek et al. (2006) estimated that the changes in aerosol size distribution explained over 80% of the observed variance in CCN concentration. Similarly, Ervens et al. (2007) stated that the CCN predictions were the most sensitive to variations in particle size distribution and supersaturation.

Considering the great climatological uncertainty related to aerosol-cloud interactions, such a straightforward relationship

between aerosol size distributions and CCN spectra could substantially improve the estimates of the aerosol indirect radiative forcing. However, as Hudson (2007) stated, the estimation by Dusek et al. (2006) was based on rather small variation in chemical composition, which may have led to considerable underestimation of its role in cloud droplet activation. In addition, Quinn et al. (2008) parametrized the aerosol chemical composition by using the mass fraction of hydrocarbon-like organic aerosol (HOA) and combined the results with CCN activity measurements. The results indicated that the uncertainty between

the measured and estimated CCN concentrations was up to 50% when the variations in HOA were neglected.

During the last few decades, a lot of effort has been put into CCN closure studies, i.e. determination of CCN concentration by means of size and composition related properties (Broekhuizen et al., 2006; Conant et al., 2004; Ervens et al., 2007; Fountoukis et al., 2007). CCN closure studies allow to assess the importance of different parameters on cloud droplet activation and therefore, they are an important tool towards a sufficient understanding of the link between atmospheric aerosols and cloud

processes. One specific method includes the determination of CCN spectra by means of hygroscopicity measurements. Under subsaturated conditions (i.e. relative humidity (RH) < 100%), aerosol hygroscopicity can be characterized by using a parameter called hygroscopic growth factor

$$GF(RH, D_p) = \frac{D_{wet}(RH, D_p)}{D_p}, \tag{1}$$

where $D_{wet}$ is the wet particle diameter at a certain RH and $D_p$ is the corresponding dry diameter. In principle, GFs reflect the

aerosol chemical composition. Pure inorganic salts such as sodium chloride (NaCl) and ammonium sulfate ($(NH_4)_2SO_4$) are usually associated with elevated growth factors (GF > 1.60 at RH = 90%), whereas some other species appear clearly less hygroscopic (McFiggans et al., 2006). For example, fresh mineral dust and pure black carbon are almost hydrophobic with typical GFs below 1.05 (Vlasenko et al., 2005; Weingartner et al., 1997; Liu et al., 2013). In atmospheric conditions, the average GFs usually vary between 1.0 and 1.8, and the GF distributions may consist of several independent modes originating

from various sources of particulate matter (Ferron et al., 2005, Fors et al., 2011; Sjogren et al., 2008; Liu et al., 2011; McFiggans et al., 2006).

In order to better understand the relationship between cloud droplet activation, aerosol chemical composition and hygroscopicity, we organized an intensive measurement campaign at the Puijo measurement station in Kuopio, Eastern Finland, during the autumn 2014 (Puijo Cloud Experiment 2014). Along with the Global Atmospheric Watch (GAW) stations such as Pallas, Finland (Komppula et al., 2005), Jungfraujoch, Switzerland (Sjogren et al., 2008) and Puy-de-Dôme, France (Asmi et al., 2012), the Puijo station is one of the few measurement sites, where long-term continuous in situ measurements on aerosol-cloud interactions are being carried out. Therefore, it provides an established base for detailed aerosol and cloud studies. Utilizing the conducted in-cloud measurements, this paper aims to identify the hygroscopicity-dependent activation properties of a cloud-forming aerosol population and study the effects of varying chemical composition on cloud droplet formation.

## 2 Methods

### 2.1 Measurement site

The measurement campaign took place at the Puijo SMEAR IV station in Kuopio, Finland (around 340 km to NE from Helsinki) between 17 September and 4 November 2014. The measurement station is located at the top of the Puijo observation tower (62°54'34'' N, 27°39'19'' E) approximately 224 m above the surrounding lake level (Leskinen et al., 2009; Ahmad et al., 2013; Portin et al., 2014). The station was established in 2005 and since then it has provided continuous data on aerosol and cloud droplet size distributions, aerosol optical properties, atmospheric trace constituents and various weather parameters. Due to the diverse surroundings, the measurement site can be characterized as a semi-urban environment. The eastern side of the tower (0–215°) includes several local pollution sources such as a paper mill in the northeast, the city center in the southeast, a heating plant in the south and the two main roads in south–north direction, as well as residential areas with occasional domestic biomass combustion. By contrast, the western sector (215–360°) has no important local sources besides the relatively small residential areas. For more detailed overview of the station and local pollution sources, see Leskinen et al. (2012) and Portin et al. (2014).

### 2.2 Weather data and cloud events

Temperature, visibility, wind speed, wind direction, precipitation, air pressure and relative humidity were measured continuously during the campaign. Visibility and precipitation were measured with a Vaisala FD12P weather sensor, wind speed and direction with an ultrasonic two-dimensional anemometer (Thies UA2D), and temperature and relative humidity with a Vaisala HMT337 temperature and humidity transmitter. Based on meteorological conditions, the measurement period was divided into cloudy and cloud-free sub-periods. Cloud events were defined as continuous periods (duration more than 1 h) with visibility below 200 m. For cloud-free conditions, the lower visibility limit was set to 8000 m in order to avoid the biasing effects of non-uniform clouds and fog. In addition, the observed events were classified as rainy if the average rain

intensity exceeded 0.2 mm h$^{-1}$. However, these cases were omitted from the analyses, since precipitation removes both activated and non-activated particles from the atmosphere causing a possible source of error for the measurement data.

## 2.3 Twin inlet system and size distribution measurements

During the campaign, we utilized a custom made twin inlet system (Portin et al., 2014) consisting of two separate sample lines in order to measure total and non-activated (interstitial) particles separately. The total air inlet had an approximate cut-off diameter of 40 μm and the sample line was heated to 30–40 °C. Due to the heating, liquid water was evaporated from the droplets and residual particles were formed. Thus, when the top of the tower was inside a cloud, the total flow contained both the residual and interstitial particles. Meanwhile, the interstitial sample line was equipped with a PM$_{1.0}$ impactor (lower cut-

off limit of 1 μm) to prevent the cloud droplets from entering the sampling system. During cloud-free conditions, both of the inlets sampled the same aerosol population.

Aerosol size distributions from 3 to 800 nm were measured with a twin differential mobility particle sizer (twin-DMPS). In this setup, two independent DMPSs measured the particle diameters from 3 to 53 nm and from 30 up to 800 nm with aerosol/sheath flow ratios of 1.4:23 and 1:5.5, respectively. The instruments were attached to the twin inlet system with an

15 automated valve, which was operated to switch between the sampling lines in 6-minute intervals. Consequently, the total measurement time for total and interstitial population was 12 minutes, assuming that the aerosol was not changing during the cycle. Full data inversion was applied to the raw data including corrections for sampling losses, multiple charging probabilities, instrumental transfer functions and particle counting efficiencies as recommended by Wiedensohler et al. (2012).

## 2.4 HTDMA experiments

Aerosol hygroscopicity under subsaturated conditions was measured with a hygroscopic tandem differential mobility analyzer (HTDMA, Table S1). Briefly, HTDMA consists of two differential mobility analyzers (DMA) and a humidifier. The first DMA (DMA1) selects the dry size of interest and the second DMA (DMA2) coupled with a condensation particle counter (CPC) measures the resulting size distribution, after the nearly monodisperse aerosol sample has been exposed to certain RH. During the campaign, the initial dry sizes were 80, 120 and 150 nm and the RH inside the DMA2 (RH$_{DMA2}$) was adjusted to

~90%. Both of the DMAs were 28 cm long Vienna type DMAs (Winklmayr et al., 1991) operated with flow ratios of 1:6, and the RH control of the sheath air inside DMA2 was achieved by using a closed-loop circulation. The residence time inside the aerosol humidifier was approximately 0.2 s, after which the particles spent ~2 s in elevated RH before entering the DMA2. The propagated instrumental uncertainty associated with the measured GFs was approximately ±4.5%.

The HTDMA was attached to the twin inlet system for a 4-week period (26 September – 20 October; hereafter referred as a

30 twin inlet period) in the middle of the campaign. Otherwise, total aerosol was being measured continuously. During the twin inlet period, an external valve system switched between the two sampling lines in 24-minute intervals. Therefore, the duration of individual size scans was adjusted so that the whole measurement cycle with three initial dry sizes was performed twice

between each line change. However, some of the data points had to be removed afterwards because the line change had occurred during an ongoing size scan. In the case of continuous total line measurements, each size scan took 5 minutes.

The average hygroscopic growth factors and their probability density functions (GF-PDF) were evaluated using the TDMAinv inversion toolkit (Gysel et al., 2009). The procedure calculates the broadening factor of the instrumental transfer function from dry size measurements and then describes the inverted GF-PDFs as piecewise linear functions. For this purpose, dry size scans with ammonium sulfate particles were performed at low RH before and during the campaign. The inversion algorithm was operated to solve the GF-PDFs with a resolution of $\Delta GF = 0.10$ and only the size scans with average $RH_{DMA2}$ between 88% and 92% were taken into account in the analysis. In addition these data points were corrected to the 90% target RH by using the built-in $\gamma$-model within the inversion toolkit. Briefly, $\gamma$-correction adjusts the measured growth factors to the desired RH by applying a parametrization $GF = (1-RH)^{-\gamma}$, where $\gamma$ is first calculated from the original measurement data and then substituted backwards to obtain the RH corrected GF.

## 2.5 Derivation of $N_{act,HTDMA}$

According to the Köhler theory (Köhler, 1936), the equilibrium saturation ratio, $S_{eq}$, over a liquid droplet can be calculated by

$$S_{eq} = a_w \exp\left(\frac{4M_w\sigma}{RT\rho_w D_{wet}}\right), \tag{2}$$

where $a_w$ is the water activity, $M_w$ the molar weight of water, $\sigma$ the surface tension (here, assumed to be that of water, 0.072 J m$^{-2}$), $R$ the universal gas constant, $T$ the ambient temperature and $\rho_w$ the density of water. In order to relate the measured growth factors to water activities, we used the $\kappa$-Köhler model described by Petters and Kreidenweis (2007). With $\kappa$-Köhler model, the water activity can be parametrized as

$$\frac{1}{a_w} = 1 + \kappa\frac{V_p}{V_w}, \tag{3}$$

where $V_p$ and $V_w$ are the dry particle and water volumes, and $\kappa$ is the hygroscopicity parameter determined as below:

$$\kappa\left(GF, D_p, RH\right) = \frac{(GF^3-1)\exp\left(\frac{4M_w\sigma}{RT\rho_w D_p GF}\right)}{RH} - GF^3 + 1. \tag{4}$$

Alternatively, the relationship between $\kappa$ and critical saturation ratio, $S_c$, i.e. the maximum of the Köhler curve (Eq. 2), can be approximated by

$$\kappa\left(D_p, S_c\right) = \frac{4A^3}{27D_p^3\ln^2 S_c}, \tag{5}$$

where $A = 4M_w\sigma/RT\rho_w$. Thus, by combining the Eqs. (4) and (5) and by assuming a certain value for $S_c$, it is possible to estimate the critical growth factor, $GF_c$, i.e. the required growth factor for particles with dry size $D_p$ to become activated at the given supersaturation. Thereafter, the size-resolved activation efficiency $f_{act,HTDMA}$ can be calculated according to

$$f_{act,HTDMA}(D_p, S_c) = \int_{GF_c(D_p,S_c)}^{\infty} GF\text{-}PDF(GF, D_p)\,dGF. \tag{6}$$

Furthermore, the available CCN concentration can be obtained by weighting the measured particle size distribution with the activation efficiency and by integrating over the whole size range:

$$N_{act,HTDMA}(S_c) = \int_{-\infty}^{\infty} f_{act,HTDMA}(D_p, S_c) \frac{dN_{tot}}{d\log D_p} d\log D_p. \tag{7}$$

In order to solve the preceding equation, the GF-PDFs determined for $D_p = 80$, 120 and 150 nm were linearly interpolated to cover the whole size range of interest. This was done by interpolating the GF-PDFs over the measurement range (80–150 nm)
and then extrapolating up to 200 nm. In addition, the 80 and 200 nm GF-PDFs were assumed to be representative for particles smaller than 80 nm and larger than 200 nm, respectively.

The procedure follows the principles described previously by e.g. Kammermann et al. (2010b) and Fors et al. (2011), except that here we did not have an independent HTDMA data point around 200–250 nm. Therefore, extrapolation towards larger sizes is also the main source of uncertainty, as underestimation of aerosol hygroscopicity could possibly lead to
underestimation of $f_{act,HTDMA}$ and vice versa. Secondly, the method assumes that the subsaturated hygroscopicities are representative for supersaturated conditions. Such an assumption is not always totally valid and discrepancies between the two saturation regimes have been reported based on laboratory and field experiments (Good et al. 2010; Hersey et al., 2014; Jaatinen et al., 2014; Pajunoja et al. 2015). Typically, the subsaturated hygroscopicities can appear somewhat lower than the supersaturated ones, which may result in underestimation of CCN concentration at a fixed supersaturation. On the other hand,
for example Jurányi et al. (2013) found a very good closure between the sub- and supersaturated regimes for an externally mixed urban aerosol in Paris, France.

In order to estimate the cloud droplet concentration in atmospherically relevant conditions, the effective peak supersaturation, $s_{c,eff}$ (where $s_{c,eff} = S_{c,eff} - 1$), was approximated by minimizing the difference between the measured and estimated activation curves, i.e. by minimizing the norm

$$\|R(S_c)\| = \sqrt{\Sigma_i [f_{act,HTDMA}(D_{p,i}, S_c) - f_{act,DMPS}(D_{p,i})]^2}, \tag{8}$$

where $D_{p,i}$ was limited to vary from 80 to 200 nm, and the DMPS derived activation efficiency, $f_{act,DMPS}$, was determined by means of total and interstitial particle size distributions as follows:

$$f_{\text{act,DMPS}}(D_{\text{p}}) = \frac{\frac{dN_{\text{tot}}}{d\log D_{\text{p}}}(D_{\text{p}}) - \frac{dN_{\text{int}}}{d\log D_{\text{p}}}(D_{\text{p}})}{\frac{dN_{\text{tot}}}{d\log D_{\text{p}}}(D_{\text{p}})}. \tag{9}$$

In principle, the effective peak supersaturation can be described as the maximum supersaturation that the particles experience for an adequate time to form stable cloud droplets (Hammer et al. 2014). It's also important to point out that this kind of approach masks the potential discrepancies between the two saturation regimes, and underestimation of supersaturated hygroscopicity would eventually lead to positive bias in $s_{\text{c,eff}}$.

In addition to the "externally mixed approach" described above, the cloud droplet concentrations were estimated under an assumption that the ambient aerosol was completely internally mixed. In this approach, the interpolated GF surfaces were used to calculate the average hygroscopicity parameter $\kappa_{\text{avg}}$ (Eq. 4) for each dry size. These values were then compared to critical $\kappa$ values approximated by using the Eq. (5) and the previously determined $S_{\text{c,eff}}$. Furthermore, depending on whether the $\kappa_{\text{avg}}$ was higher or lower than the respective critical value, the activation efficiency was assumed to be either 1 or 0, respectively. Therefore, this kind of internally mixed approach resulted in activation efficiency curves resembling a step function instead of S-shaped curves obtained by means of full GF-PDFs.

# 3 Results and discussion

## 3.1 Weather parameters and cloud events

During the campaign, the hourly averaged ambient temperatures varied between -9.7 °C and 15.3 °C, with sub-zero temperatures occurring mostly during the latter half of the measurement period. Due to the diurnal temperature variations, relative humidity usually reached its daily maxima during the nighttime and early morning hours. Therefore, most of the cloud formation events also occurred within these time periods. Generally, the wind speeds exhibited a diurnal variation similar to ambient RH with slightly increased velocities during the nighttime. The wind was blowing from southwestern directions (180–270°) approximately 39% of time, and during cloud events this fraction was even higher, roughly 50%. Altogether, the tower was inside a cloud approximately 10% of time, which is somewhat less compared to typical autumn conditions at Puijo (Portin et al., 2009). In total, 15 non-precipitating cloud events were observed during the campaign, providing up to 47 cloud event hours.

## 3.2 Hygroscopicity at Puijo

Figure 1 shows the average GFs and GF-PDFs observed at Puijo during the campaign period. Overall, the measured GFs varied between ~1 and ~1.7 reaching the campaign averages (± 1 standard deviation) of 1.30 ± 0.11 (80 nm), 1.42 ± 0.10 (120 nm) and 1.47 ± 0.10 (150 nm). A closer look at the GF-PDFs indicates that there are two main factors affecting the observed size dependence. In general, the GF-PDFs appear bimodal with clearly distinguishable peaks around 1.0–1.1 and 1.4–1.6. However, the number fraction of less hygroscopic particles ($f_{\text{GF}<1.25}$) decreases with increasing particle size, and at the same time, the

more hygroscopic mode tends to shift slightly towards larger growth factors. A similar shift is also apparent in Fig. 2 where the average hygroscopicity distributions are plotted as $\kappa$-PDFs. Furthermore, this indicates that the transition towards higher GFs can be attributed both to Kelvin effect as well as changes in chemical composition.

According to a wind sector analysis (see supplementary material for further details), the polluted sector was characterized by
higher $f_{GF<1.25}$ than the clean one, but on the other hand, the more hygroscopic mode shifted towards higher GFs suppressing the differences in average GFs. Tiitta et al. (2010) studied the particulate roadside emissions in Kuopio and observed an elevated non-hygroscopic particle mode ($D_p \leq 50$ nm) originating from fresh exhaust emissions. A similar observation was also made by Laborde et al. (2013) in Paris with particle sizes up to 265 nm. Against this background, it is probable that the less hygroscopic fraction observed at Puijo is also linked to the traffic and other anthropogenic sources of organic aerosol.

Minor differences can be seen also in the GF-PDFs between cloudy and cloud-free conditions (Fig. 1). The fraction of more hygroscopic particles is slightly enhanced during cloud-free conditions, resulting in higher average GFs. This can be due to the limited time scale of cloud events and varying contribution of local pollution, together with possibility for in-cloud processing and precipitation scavenging. However, for example Henning et al. (2014) and Zelenyuk et al. (2010) reported a cloud-induced sulfate enrichment in the particle phase, which should eventually result in increasing hygroscopicity. One may
also note that the in-cloud distributions measured during the twin inlet period differ considerably from the campaign averages with substantially stronger contribution of $f_{GF<1.25}$ at $D_p = 120$ and 150 nm. Again, the most probable explanation is the enhanced presence of local pollution due to more frequent easterly winds.

As indicated above, the analytical division between the less and more hygroscopic particles was done by using a GF limit of 1.25. The value represents a common midpoint between the two GF modes and it has been used in several preceding studies
(e.g. Kammermann et al., 2010a; Jurányi et al., 2013; Portin et al., 2014). Averaged over the whole campaign, the size-dependent less hygroscopic fractions were $0.44 \pm 0.25$ (80 nm), $0.22 \pm 0.17$ (120 nm) and $0.15 \pm 0.14$ (150 nm). Together with the average GF-PDFs, these values are comparable to hygroscopicities observed at a back-ground site in southern Sweden (Fors et al., 2011). However, as seen during the twin inlet period, our site experienced reasonably strong variation in less hygroscopic fraction with $f_{GF<1.25}$ regularly reaching the levels of urban conditions (McFiggans et al., 2006). For example during
the observed cloud hours, the average $f_{GF<1.25}$ ranged from 0 up to 0.62 at $D_p = 150$ nm. By comparison, Jurányi et al. (2013) studied the aerosol mixing state in Paris, France and reported typical values varying between ~0.20 and ~0.60 at $D_p = 165$ nm. In addition, Ferron et al. (2005) observed that in urban and semi-urban conditions, the non-hygroscopic mode could be dominant at particle sizes up to 250 nm, especially during the autumn and winter seasons.

### 3.3 Hygroscopic properties of total, interstitial and residual aerosol

The GF-dependent activation efficiencies ($f_{act,GF}$) were derived from the measured GF-PDFs according to

$$f_{act,GF_1<GF<GF_2}(D_p) = \frac{\frac{dN_{tot}}{d\log D_p}(D_p) \times f_{tot,GF_1<GF<GF_2}(D_p) - \frac{dN_{int}}{d\log D_p}(D_p) \times f_{int,GF_1<GF<GF_2}(D_p)}{\frac{dN_{tot}}{d\log D_p}(D_p) \times f_{tot,GF_1<GF<GF_2}(D_p)}, \qquad (10)$$

where $f_{tot,GF1<GF<GF2}$ and $f_{int,GF1<GF<GF2}$ were the total and interstitial number fractions of particles with dry size $D_p$ and GF between $GF_1$ and $GF_2$. The average activation efficiencies were calculated separately for each cloud event and for three different GF regimes ($GF \geq 0.80$, $0.80 \leq GF < 1.25$ and $GF \geq 1.25$). Here it should be noted that the activation efficiencies can appear negative if the averaged interstitial concentrations are higher than the corresponding total values. This can be the case especially within the less hygroscopic regime where the activated fractions are generally low. In such cases, the negative activation efficiencies are reported as zeros and treated as such when calculating the total activated fractions ($f_{act,GF \geq 0.80}$). Thus, the resulting $f_{act,GF \geq 0.80}$ values can be slightly different from those derived solely from DMPS measurements.

A total of nine cloud events were observed during the twin inlet period. Due to the reasonably low time resolution of HTDMA size scans and slow alteration between the two sampling lines, data with good coverage and reasonable agreement between $f_{act,GF \geq 0.80}$ and $f_{act,DMPS}$ were available for four separate cloud events. These cases are summarized in Table 1. Because of the low activation efficiency of ultrafine particles, the activation parameters are presented only for 120 and 150 nm sizes.

The duration of the selected cloud events varied from 1 h 31 min (event #2) to 4 h 25 min (event #4). Cloud event #1 had the highest particle and cloud droplet number concentrations, up to 2935 cm$^{-3}$ and 781 cm$^{-3}$, respectively, whereas much lower values were observed during the latter events. For example during the cloud event #4, the particle and cloud droplet concentrations were down to 792 cm$^{-3}$ and 69 cm$^{-3}$, respectively. These four cloud events were also characterized by very different wind patterns. Event #2 was influenced by westerly winds blowing across the clean sector. By contrast, the wind was from the northeast during cloud event #3 and from the southeast during the cloud event #4. Furthermore, event #1 was dominated by southwesterly winds blowing across the transition region between the clean and polluted sectors.

The most interesting remark, however, concerns the different activated fractions within the two GF modes. At $D_p = 120$ nm, the activation efficiencies of less hygroscopic particles varied from 0% to 4%, whereas the values for more hygroscopic particles were much higher (between 57% and 70%). A similar trend was observed also at $D_p = 150$ nm, with corresponding intervals of 0%–34% and 78%–83%, respectively.

One may also note that the hygroscopicity-dependent activation efficiencies increased with particle size. In the case of more hygroscopic particles, this was most likely attributed to Kelvin effect and small increments in respective hygroscopicities. Besides, the cloud events #2 and #4 were characterized by somewhat increased $f_{act,GF<1.25}$ values at $D_p = 150$ nm. Although it is possible that part of these less hygroscopic particles were scavenged into cloud droplets, it's good to note that these cases were also characterized by notably low particle concentrations which may have led to increased uncertainties in activated fractions.

The difference between the activated and non-activated particles is also illustrated in Fig. 1 (lower panel) where the average GFs and GF-PDFs are presented separately for total, interstitial and residual aerosol populations. Here, the residual aerosol properties were estimated indirectly by using the hourly averaged total and interstitial GF-PDFs and the respective ambient particle concentrations as follows:

$$c_{res}(GF, D_p) = \frac{dN_{tot}}{d \log D_p}(D_p) \times GF\text{-}PDF_{tot}(GF, D_p) - \frac{dN_{int}}{d \log D_p}(D_p) \times GF\text{-}PDF_{int}(GF, D_p). \tag{11}$$

Apparently, the relative contribution of less hygroscopic particles is the strongest in the interstitial population and the more hygroscopic mode appears distinctly only in the total and estimated residual aerosol. This is also reflected by the average growth factors. At $D_p$ = 120 and 150 nm, the GFs of cloud droplet residuals are approximately 18% higher than those of interstitial particles. In terms of $\kappa$ values, this discrepancy would correspond to a difference up to 65%–75% depending on

particle size. Again, it should be remarked that similar to activated fractions, the residual GF distributions ($c_{res}$) can appear locally negative if the interstitial concentrations are higher than the respective total concentrations. Before converting these distributions into GF-PDFs shown in Fig. 1, the negative values were set to zeros which eventually strengthened the positive peaks appearing in normalized distributions. As a result, for example the less hygroscopic particle mode appearing in the estimated residual aerosol can be partially attributed to methodological uncertainties.

To our knowledge, this is one of the very few studies characterizing the hygroscopic properties of different in-cloud aerosol populations. Previously, Svenningsson et al. (1994) studied the aerosol hygroscopicity and its relationship to cloud droplet activation at Kleiner Feldberg in Germany. Interstitial aerosol hygroscopicity was measured during cloud events, and by assuming that the air mass was the same, it was compared to the total aerosol sampled during the following clear sky conditions. The less hygroscopic particle fraction was substantially higher in the interstitial population, indicating that the more

hygroscopic particles were scavenged into the cloud droplets more efficiently than the less hygroscopic ones. This observation was confirmed in a case study, where a counter flow virtual impactor (CVI) was used to separate the cloud droplets from total aerosol, so that the hygroscopicity of cloud droplet residuals could be measured independently. Likewise, Rose et al. (2013) observed a decreasing fraction of available CCN during a precipitative cloud event, suggesting that the more hygroscopic particles were mostly activated into cloud droplets and removed from the air through precipitation. Overall, our results from

Puijo are in good agreement with the observations reported by Svenningsson et al. (1994) and Rose et al. (2013).

In situ aerosol chemical composition and partitioning between activated and non-activated particles has been studied recently by means of aerosol mass spectrometry (e.g. Hao et al., 2013; Zelenyuk at al., 2010; Kamphus et al., 2010; Drewnick et al., 2006) and electron microscopy (Li et al., 2011). Based on the results from Puijo Cloud Experiment 2010, Hao et al. (2013) reported substantially higher mass fractions of organic nitrate and less oxidized organic species in the interstitial particles

compared to residual ones. Similarly, Zelenyuk et al. (2010) and Li et al. (2011) observed an elevated fraction of sulfates in the activated particles. Although these results are broadly in line with our observations on the distinct hygroscopicity of activated and non-activated particles, great care should be taken when comparing the results. For example, Hao et al. (2013) observed an increased organic fraction in the Aitken mode particles, indicating that the differences in chemical composition also reflected the effect of particle size.

At Puijo, the critical droplet activation diameter, i.e. the diameter corresponding to the 50% activation efficiency (D50), varies typically between 100 and 200 nm. According to our observations, this value is highly dependent on the prevailing hygroscopicity. To illustrate this relationship, the DMPS derived D50s (obtained from $f_{act,DMPS}$ via linear interpolation) were correlated with hourly averaged $\kappa$ values. Furthermore, a non-linear regression model D50 $\sim$ a $\times$ $\kappa^{-\frac{1}{3}}$ was chosen to account for the theoretical dependence addressed in Eq. (5). Figure 3 summarizes the results from all the observed cloud hours. Since the

average hygroscopicity reflects the relative contribution of the two GF modes, the low-end $\kappa$ values were characterized by elevated $f_{GF<1.25}$ and vice versa. Differing more than three standard deviations from the campaign average of 132 nm, the data point with D50 = 231 nm was excluded from the analysis as an outlier. Apart from this exception, the hourly D50s ranged from 93 to 173 nm.

The 80 nm particles yielded the weakest correlation with root-mean-square error (RMSE) of ~29 nm. This could be expected since the 80 nm size usually remained well below the critical activation limit. On the contrary, the 120 and 150 nm sizes yielded the RMSE values around ~13 nm, suggesting a considerably stronger correlation between the two parameters. According to the linear correlations reported by Quinn et al. (2008), the chemical composition (parametrized as HOA mass fraction) explained approximately 40% to 50% of the variation in critical activation diameter. In comparison, applying a linear

regression to our data set would result in $R^2$ values of 0.58 (120 nm) and 0.57 (150 nm). Assuming that the relationship between $\kappa$ and D50 could be treated as (locally) linear, these values would indicate that the accumulation mode hygroscopicity explained up to 57%–58% of the observed variance in D50, therefore dominating the effect of varying meteorology and especially, the varying supersaturation. Moreover, no correlation was found between the $\kappa$ values and the estimated effective peak supersaturations ($R^2 \sim 0.02$; not shown here).

**3.4 $N_{act,HTDMA}$ and $s_{c,eff}$**

The activation efficiency curves and the corresponding cloud droplet number concentrations were derived from the hourly averaged GF-PDFs according to the procedure described in Sect. 2.5. Reproducibility of the DMPS derived activation curves and cloud droplet nuclei spectra was confirmed visually, and the data points with considerable differences were omitted from the following analyses. In most of these cases, the estimated activation curves appeared clearly steeper than the measured ones,

leading to both an underestimation of activated fraction at $D_p < $ D50 and an overestimation of activated fraction at $D_p > $ D50. Alternatively, some of the estimated curves managed to reproduce the correct behavior at smaller sizes ($D_p < 150$ nm) but overestimated the activation at the size range of extrapolated GF-PDFs. Although these uncertainties caused only a small net bias to droplet estimations, they indicate that the estimated GF surfaces may have failed to replicate the real ambient conditions. Figure 4 shows a comparison between the HTDMA and DMPS derived cloud droplet concentrations (red dots). The regression

line fitted through the 26 data points has a slope of 1.016 indicating a good agreement between the estimated and measured cloud droplet concentrations. In addition, Fig. 4 includes a comparison between the external and the internal mixing approaches (grey dots). Correspondingly, the regression line has a slope of 1.022 and all the hourly data points lie between the ratios 0.91 and 1.04. Analogous to Kammermann et al. (2010b), this observation suggests that the CCN concentration can be determined with reasonable accuracy even if the exact mixing state remains unknown. Nevertheless, since the determination of $N_{act,HTDMA}$

included the estimation of effective peak supersaturation by means of $f_{act,DMPS}$, the comparison shown here can't be considered as a real CCN closure study in an explicit manner. Instead, it rather serves as a base for the analysis presented in the following section (Sect. 3.5).

Overall, the hourly estimated peak supersaturations ranged from 0.16% to 0.29%. Apart of one exception with $s_{c,eff}$ up to 0.44%, this range was valid also for the omitted cloud hours. Although these values might be slightly biased due to the possible discrepancies between sub- and supersaturated hygroscopicities, they provide valuable information about the conditions relevant to cloud droplet formation. Reaching the average and median values of 0.22%, the estimated $s_{c,eff}$ values are comparable to supersaturations of 0.18%–0.26% obtained by Anttila et al. (2009) at the Pallas GAW station in Northern Finland. Similar "average" supersaturations can be obtained also by using the regression curves presented in Fig. 3. By assuming a constant temperature of 2 °C (average temperature of all the cloud hours), the regression parameters $6.94 \times 10^{-8}$ (120 nm) and $7.52 \times 10^{-8}$ (150 nm) imply supersaturations of 0.23% and 0.20%, respectively. By contrast, these values are somewhat low compared to the high-end supersaturations determined at some European mountain sites. For example, Asmi et al. (2012) found supersaturations ranging from 0.1% up to 0.6% at Puy-de-Dôme, and similarly, Hammer et al. (2014) observed a median peak supersaturation of 0.35% at Jungfraujoch. On the other hand, Hammer et al. (2014) also found a clear difference between the two dominant wind sectors reflecting the effect of terrain topography on updraft velocities. Generally, the air masses rising over the less steep mountainside were characterized by weaker updrafts and consequently, by supersaturations more comparable to ours (median 0.22%).

**3.5 Sensitivity of cloud droplet formation to varying hygroscopicity**

Following the procedure described in Sect. 2.5, we performed $\kappa$-Köhler simulations to investigate how the cloud droplet concentrations would change if all the particles belonged to the more hygroscopic group. For each cloud hour, we created an alternative hygroscopicity scenario by modifying the original GF-PDFs. This alteration was conducted by eliminating the contribution of less hygroscopic particles and normalizing the resulting distributions so that the integral over each particle size became equal to one. In principle, the most significant changes appeared at sizes where the original $f_{GF<1.25}$ were large enough to be affected by the modifications. Thus, instead of varying the GFs equally and regardless of particle size, the modifications only affected the presence of an existing low-GF mode and increased the average GFs accordingly.

Overview of the applied hygroscopic variations is presented in Table 2. Shown are average GFs at $D_p$ = 80, 120, 150 and 200 nm for the two hygroscopicity scenarios, as well as their absolute deviation from each other. In terms of $\kappa$ values, the high hygroscopicity assumption results in values between 0.20 and 0.40. According to Andreae and Rosenfeld (2008), these values would be characteristic of typical aged continental aerosol. In order to highlight their atmospheric relevance, it's also important to note that they are within the range of hygroscopicities observed at Puijo during the campaign period. The cloud droplet concentrations were calculated for both scenarios by using identical particle size distributions as well as equal effective peak supersaturations.

Figure 5 shows the relative and absolute changes in $N_{act,HTDMA}$ against the hygroscopicity shift at $D_p$ = 150 nm. Again, the data points are colored according to the less hygroscopic fraction, and the marker size is scaled relative to total particle number concentration. Naturally, the relative change in cloud droplet concentration increases with increasing hygroscopicity. Although most of the data points reside in the range of 10%–40%, the total variation extends up to 70%. Furthermore, because of the

approximately linear behavior of relative $\Delta N_{\mathrm{act,HTDMA}}$, the absolute change appears highly sensitive to initial particle size distribution. In the case of high total concentration, the absolute $\Delta N_{\mathrm{act,HTDMA}}$ can be up to hundreds of droplets per cubic centimeter even with reasonably small hygroscopic variations ($\Delta GF_{150\ \mathrm{nm}} < 0.10$).

In addition to droplet concentration, Fig. 5 illustrates the change in critical activation diameter (D50$_{\mathrm{HTDMA}}$; derived from $f_{\mathrm{act,HTDMA}}$ via linear interpolation). Understandably, as the high hygroscopicity assumption suppresses the size-dependence of aerosol hygroscopicity (see Table 2), the activation efficiency curves become steeper and D50s decrease. Typically, the change in D50 remains below 20 nm but can reach 35 nm in the most extreme cases.

Anttila et al. (2009) studied the effect of varying hygroscopicity at the Pallas GAW station in Northern Finland. According to the simulations, increasing the GFs by 10% led to 17%–51% increments in cloud droplet number concentration. Bearing in mind that a 10% relative change in GFs corresponded to an absolute change of ~0.13 at $D_{\mathrm{p}} = 150$ nm, the observations by Anttila et al. (2009) are in good agreement with our results. However, it is also worth pointing out that by increasing all the GFs by 10%, the absolute change in growth factors increased with particle size. In our analysis, removal of the less hygroscopic mode induced the opposite effect.

Similar to our analysis, Wex et al. (2010) assessed the overestimation of CCN concentration if all the particles were assumed to have $\kappa$ of the more hygroscopic particles. By using data from the literature, the overestimation was derived for rural, urban and marine aerosol populations and for different less hygroscopic fractions. In the case of urban and rural aerosols, the overestimation varied between 20%–40% and 40%–100% when the less hygroscopic fraction ($\kappa < 0.10$) was set to 0.3 and 0.5, respectively. With the range of variation illustrating the inverse proportionality to supersaturation (between 0.1% and 0.5%), these values are comparable to our observations. However, Wex et al. (2010) assumed the same less hygroscopic fraction for all particle sizes, which is very rarely the case in our conditions at Puijo.

Kammermann et al. (2010b) performed a set of CCN closure studies to investigate the sensitivity of CCN concentration to unknown chemical composition. The median bias between the predicted and measured CCN concentrations was up to +54% when a constant $\kappa = 0.30$ was assumed. Analogous to our observations, the average hygroscopicities reported by Kammermann et al. (2010b) were relatively low explaining the remarkable bias in CCN predictions. On the contrary, Meng et al. (2014) found a reasonably good agreement between the measured and predicted CCN concentrations by using a constant $\kappa$ as high as 0.33 at a coastal site in Hong Kong, where the aerosol composition was dominated by inorganic species.

In principle, the high hygroscopicity assumption yields values resembling the hygroscopicities that could be obtained from bulk composition measurements by means of aerosol mass spectrometry. Typically, the bulk composition is biased towards the inorganics due to the emphasis on larger particles. This can lead to a considerable overestimation of CCN concentration. For example, Medina et al. (2007), Almeida et al. (2013) and Meng et al. (2014) reported an overprediction of 26%–44% at ~0.20% supersaturation when only the size-averaged composition was considered. In all of these studies, implementing the size-dependent or size-resolved chemical composition substantially improved the CCN predictions.

# 4 Summary and conclusions

The relationship between aerosol hygroscopicity and cloud droplet activation was studied at the Puijo measurement station in Kuopio, Finland during the Puijo Cloud Experiment 2014. The purpose of the campaign was to identify the hygroscopicity-dependent activation properties of a cloud forming aerosol population, as well as to study the sensitivity of cloud droplet activation to varying chemical composition in real atmospheric in-cloud conditions. In total, 15 non-precipitating cloud events were observed during the 2-month long campaign providing a total of 47 cloud hours.

The aerosol hygroscopicity at 90% RH was measured with an HTDMA. Typically, the measured GF-PDFs appeared bimodal, indicating an externally mixed aerosol population. By using the GF-PDFs and particle concentrations measured separately for interstitial and total aerosol populations, the hygroscopicity-dependent activation properties were estimated. The growth factor distributions were divided into low and high hygroscopicity regimes by using a GF limit of 1.25 and the activated fraction in each category was calculated.

The in-cloud measurements revealed clear differences in activation efficiency between the two GF modes. The less hygroscopic particles originating most likely from local anthropogenic sources remained mostly non-activated, whereas the more hygroscopic mode was primarily scavenged into cloud droplets. This observation highlights the role of aerosol hygroscopicity and chemical composition in cloud droplet activation. A highly variable portion of less hygroscopic particles has been observed in several studies and in many different locations during the last few decades. Due to the anthropogenic contribution, the less hygroscopic mode can be dominant at particle sizes up to 250 nm. As shown by our analysis, this can lead to a significant decrease in the fraction of available CCN.

By modifying the measured GF-PDFs, we estimated how the cloud droplet concentrations would change if all the particles belonged to the more hygroscopic mode. This would correspond to a situation with typical aged, continental aerosol in the atmosphere without any fresh anthropogenic influence. According to the $\kappa$-Köhler simulations, the change in cloud droplet concentration was up to 70% when the possible feedback effects on cloud supersaturation were assumed negligible. Our result clearly demonstrates the importance of correct treatment of anthropogenic organic aerosols, their hygroscopicity and the effect of atmospheric aging, when estimating CCN concentrations.

**Acknowledgements.** This work was supported by the European Research Council (ERC starting grant 335478), the Academy of Finland (grant no. 272041, 259005, 283031) and The Doctoral School of the University of Eastern Finland. In addition, A.R. acknowledges the financial support from Maj and Tor Nessling Foundation.

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

**Table 1: Overview of the four cloud events observed during the measurement campaign. Given are the total particle and cloud droplet number concentrations as well as the hygroscopicity-dependent activation efficiencies and growth factors of 120 and 150 nm particles. The number fraction of less hygroscopic particles is denoted by $f_{GF<1.25}$.**

| | Event #1 | | Event #2 | | Event #3 | | Event #4 | |
|---|---|---|---|---|---|---|---|---|
| Time | 08.10/20:46–23:45 | | 11.10/20:25–21:56 | | 13.10/05:57–07:40 | | 20.10/00:01–04:26 | |
| $N_{tot}$ | 2935 cm$^{-3}$ | | 699 cm$^{-3}$ | | 1442 cm$^{-3}$ | | 792 cm$^{-3}$ | |
| $N_{act}$ | 781 cm$^{-3}$ | | 135 cm$^{-3}$ | | 158 cm$^{-3}$ | | 69 cm$^{-3}$ | |
| | 120 nm | 150 nm | 120 nm | 150 nm | 120 nm | 150 nm | 120 nm | 150 nm |
| $f_{act,DMPS}$ | 0.20 | 0.40 | 0.29 | 0.55 | 0.37 | 0.68 | 0.39 | 0.68 |
| $f_{act,GF\geq0.80}$ | 0.32 | 0.48 | 0.29 | 0.55 | 0.52 | 0.72 | 0.39 | 0.68 |
| $f_{act,GF<1.25}$ | 0 | 0 | 0 | 0.31 | 0 | 0 | 0.04 | 0.34 |
| $f_{act,GF\geq1.25}$ | 0.57 | 0.80 | 0.69 | 0.77 | 0.60 | 0.79 | 0.70 | 0.83 |
| $GF_{avg,GF\geq0.80}$ | 1.27 | 1.31 | 1.20 | 1.25 | 1.58 | 1.65 | 1.27 | 1.41 |
| $GF_{avg,GF<1.25}$ | 1.13 | 1.13 | 1.05 | 1.06 | 1.00 | 1.00 | 1.01 | 1.02 |
| $GF_{avg,GF\geq1.25}$ | 1.38 | 1.42 | 1.41 | 1.42 | 1.68 | 1.71 | 1.50 | 1.58 |
| $f_{GF<1.25}$ | 0.44 | 0.39 | 0.58 | 0.47 | 0.14 | 0.08 | 0.48 | 0.31 |

**Table 2: Average GFs for reference and high hygroscopicity scenarios and their absolute deviation from each other. The values shown in the parentheses correspond to the minimum and maximum values. The GFs for $D_p$ = 80, 120 and 150 nm originate from direct measurements, whereas the data for $D_p$ = 200 nm is obtained via extrapolation.**

|  | 80 nm | 120 nm | 150 nm | 200 nm |
|---|---|---|---|---|
| $GF_{avg}$ [Reference] | 1.20 (1.04/1.29) | 1.30 (1.18/1.43) | 1.38 (1.25/1.49) | 1.44 (1.27/1.58) |
| $GF_{avg}$ [High-GF] | 1.40 (1.33/1.59) | 1.43 (1.36/1.57) | 1.47 (1.39/1.59) | 1.49 (1.4/1.61) |
| $\Delta GF_{avg}$ | 0.20 (0.07/0.43) | 0.13 (0.06/0.24) | 0.09 (0.04/0.21) | 0.05 (0/0.15) |

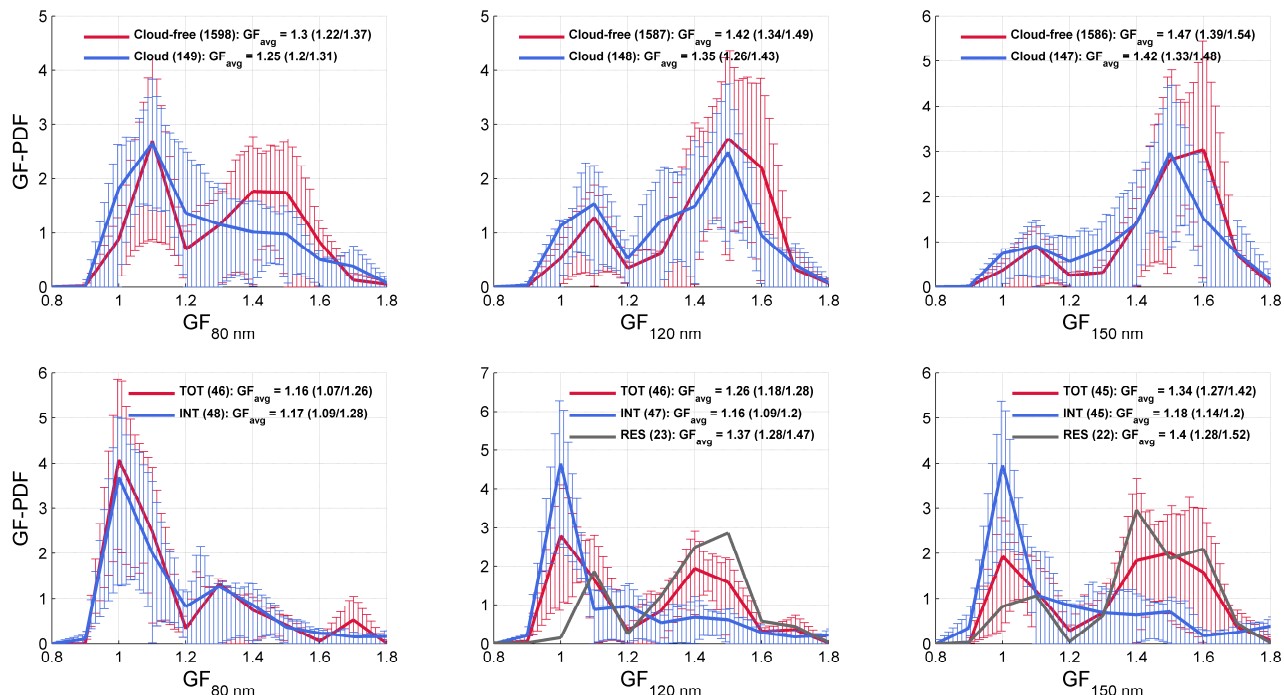

**Figure 1: Top row: Average hygroscopicity of total aerosol at 90% RH during cloudy (blue) and cloud-free (red) conditions (whole campaign). Bottom row: Average in-cloud hygroscopicity of total (red), interstitial (blue) and residual aerosol (black) during the twin inlet period. The values shown in the parentheses represent the number of averaged observations and the 25th and 75th percentiles. In the graphs, the lower and upper quartiles are illustrated with whiskers.**

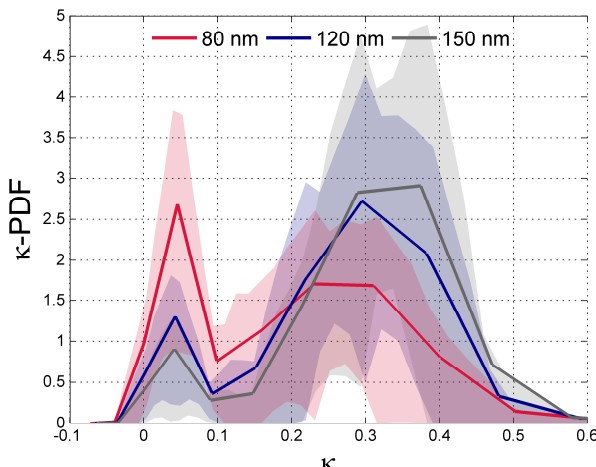

**Figure 2: Mean κ-PDFs of 80, 120 and 150 nm particles calculated over the whole data set. The shaded areas represent the ranges between the 25th and 75th percentiles.**

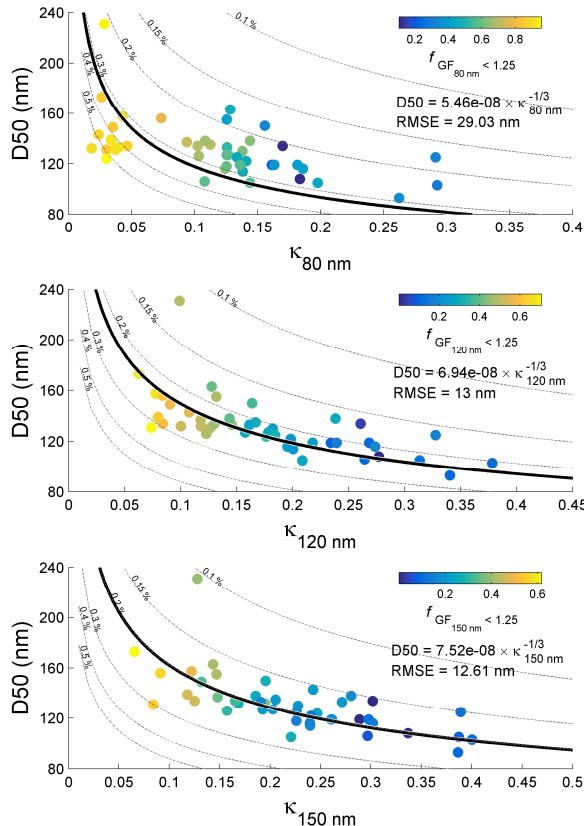

**Figure 3: Critical activation diameter (D50) vs. size-averaged hygroscopicity ($\kappa$) of 80 nm, 120 nm and 150 nm particles, as well as the power law fits (black lines) to the data. The data points are colored according to number fraction of less hygroscopic particles, and the grey dashed lines represent the numerical solutions of $\kappa$-Köhler theory in the range of $s_c$ = 0.1%–0.5%.**

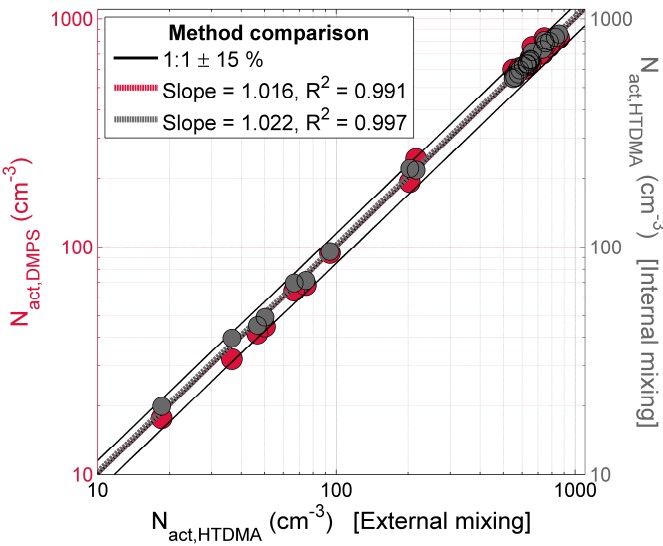

**Figure 4: Correlations between the estimated (HTDMA) and measured (DMPS) cloud droplet concentrations (red dots) and the external and internal mixing approaches (grey dots). Each data point represents an hourly average.**

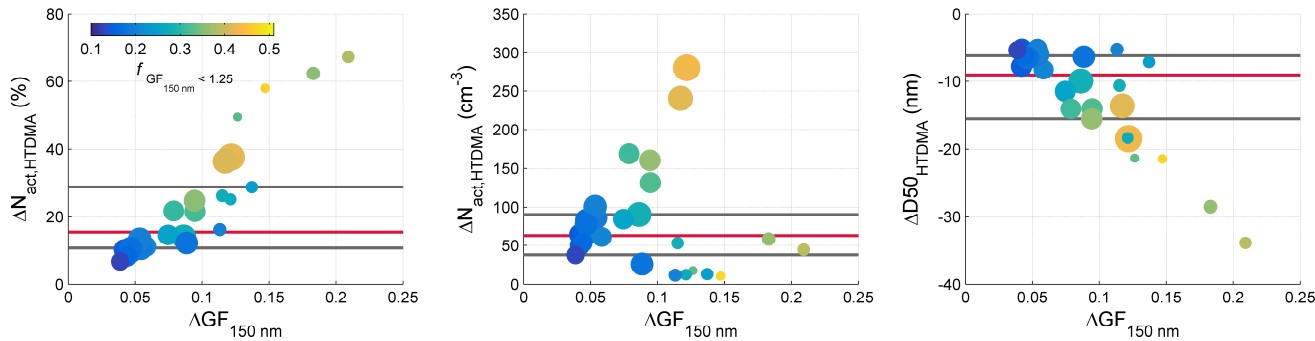

**Figure 5: Simulated changes in cloud droplet number concentration ($N_{act,HTDMA}$) and critical activation diameter (D50$_{HTDMA}$) if all the particles belonged to the more hygroscopic mode. The marker size illustrates the total particle concentration in the range of 388 cm$^{-3}$ to 3316 cm$^{-3}$ and the data points are colored according to the less hygroscopic fraction (i.e. the fraction of particles merged into the more hygroscopic mode). The horizontal lines correspond to the 25th, 50th and 75th percentiles.**