# Peer review of "In-cloud measurements highlight the role of aerosol hygroscopicity in cloud droplet formation"

_Atmospheric Chemistry and Physics, 2016_

## Referee Comment (RC1) · Anonymous Referee #1 · 22 May 2016

General comments

The manuscript "In-cloud measurements highlight the role of aerosol hygroscopicity in cloud droplet formation" investigated the hygroscopic growth distribution of particles at three sizes in cloud events and the hygroscopicity-dependent droplet activation at a tower station in Puijo, Finland. The authors found the bimodal hygroscopic growth factor distribution. This study further examined the changes of droplet activation properties if the less hygroscopic particle fractions have the same hygroscopicity as the higher hygroscopic fractions. By this, the manuscript highlighted the important role of chemical composition, namely hygroscopicity, in cloud droplet activation. This manuscript is generally well written and provides an evaluable case study on the role of chemical composition in cloud droplet activation. Together with other previous studies, it helps understand the role of size and chemical composition in CCN activation in a balanced way. Realizing that both size and chemical composition are important in CCN activation rather than over-stressing one over the other benefit the CCN studies. But before it is published on ACP, the manuscript need address the following questions.

General comments

1.  Some important details need further clarification. For example, in Pg. 4, lines 14. The authors stated that " full data inversion was applied…, including the corrections for sampling losses, multiple charge probabilities, instrument transfer function and particle count effeciencies". The corrections are important for the data evalution, but it is not clear for readers. These details should be provided in the Supplement or at minimum be referred to specific references in a detailed way.  Another example is that how the D50 in Fig. 2 and D50 in Fig. 4 were exactly derived is missing.  And "$f_{act,DMPS}(Dp)$" in Eq. 8 was not defined. More examples are included in the "specific comments" part.
2.  The authors attributed the "less hygroscopic" particles to be "originating from local anthropogenic sources". In the conclusion, it was stated that "our result clearly demonstrates the importance of correct treatment of anthropogenic aerosols, their hygroscopicity and the effect of atmospheric aging, when estimating the CCN concentration." I guess the authors specially meant the anthropogenic organic aerosol and carbonaceous aerosol since the $(NH4)2SO4$ or $NH4NO3$ are highly hygroscopic. While I agree it is important to correctly treat aerosols with both low and high hygroscopicity, it should also be noted that the biogenic secondary organic aerosols can also have a low kappa(e.g around 0. 1 as in Fig. 2). They could also contribute to the total particles even when the winds were from "anthropogenic" sectors since air masses could pass over the forest regions before reaching the measurement site(a backward trajectory may help clarify whether this could be true). Moreover, without detailed aerosol chemical composition for source apportion, it is only a plausible speculation here that the aerosol is "originating from local anthropogenic emission. I suggest the authors to use more precise wording here in the conclusion and in the abstract.

Specific comments

1.  Pg. 1, line 13, as mentioned as above, "reflecting the varying presence of fresh anthropogenic particle emission" is based on plausible speculation. Some hedge words are recommended here.

2. Pg. 4, line 22, please define the "Vienna type DMA".

3. Sect. 2.4, What is the residence time of the particles in the humidifier?

4. Pg. 6, line 22 Eq. 8, as mentioned above, "$f_{act,DMPS}(Dp)$" was not defined.

5. Pg. 8, line 1, "…during the changing inlet period…", if I understand correctly, the inlet alternated every 6 min between the total aerosol and interstatial aerosol measurement. By using "changing inlet period", did the authors suggest there were some periods when inlet was not changing? Please clarify.

6. Pg. 9 lines 4-5, "… the less hgyroscopic particle mode remains almost non-activated" is not fully convincing to me. From event #1 and # 3, some fractions (12-33%) of the particles still activated. What about other clouds events in the 15 events? The activated fractions are typically around 0 or as in event #2 or more like event #1 and #3? Maybe re-phase this sentence, saying something like "the activated fraction is much lower(value x-value y%)".

7. A related question to the last question. In Fig. 1, what are the unit and value of y-axis in these graphs? I suppose the the value is the number concentration of particle with given GF value. If I understand correctly, in the bottle middle panel of Fig.1, the residue concentration divided by total concentration would yield the activated fraction as a function of GF. Integrating the residue concentration for the range of GF<1.25 divided by the integrated total concentration would yield the activated fraction of particle with GF<1.25 ($f_{act,GF<1.25}$). It seems that the values derived in this way would not be close to zero for either 120 nm particles or 150 nm particles. Could the authors explain this?

8. Pg. 9, line 32, "… in line with the hygroscopic discrepancies…". It is not clear for me what the authors meant by "discrepancies". Do they mean the different activated fraction between the less and more hygroscopic particles? If so, please phrase it precisely.

9. Pg. 10, lines 15-17, "…hygroscopicity of accumulation mode particles explained up to 57-58% of the observed variance in D50,…especially, the varying supersaturation"., from Fig. 2, the middle and bottle panel, most data points are converged to be close to the line of $D50=Ak^{-1/3}$. Does this indicate that the variability of the supersaturation is quite small? And do these graphs include all the data from 15 clouds events?

10. Pg. 10, lines 20-21, the authors stated that "… the data points with considerable uncertainties were omitted…". Could the authors precisely describe what data were classified as ones "with considerable uncertainties" since how the data were omit would affect the correlation in Fig. 3.
    Also does the "cloud droplet nuclei spectra" refer to the spectra derived from DMPS? Please clarify.

11. Pg. 13, line 4, as mentioned above, "the less hygroscopic particles originating from local anthropogenic emissions" is only a speculation. I suggest to add a hedge word here, e.g. likely.

Technical comments

1. Pg 4, line 11, ' The instrument was…" should be "instruments were".
2. Pg 5, line 1, "…using the TDMAinversion..",  a space is missing after TDMA.
3. Pg. 10, line 2, "… the above results…" should be "…the results above…".
4. Pg. 12, line 1 , "… the absolute change in hygroscopicity increased…", is "hygroscopicity" the right word? Or growth factor?

5. Pg. 12, line 12, "Similarly to our measurement…" should be "similar".

---

## Referee Comment (RC2) · Anonymous Referee #2 · 25 May 2016

This paper analysis the influence of aerosol particle hygroscopicity on cloud droplet formation, based on field measurements. As pointed out by the authors, there are very few measurements dealing directly with this issue. The issue is at the same time relevant for aerosol-cloud-climate interactions as well as for the transport and lifetimes of air quality relevant pollutants and biogenic compounds in the aerosol. The measurements, data analysis and presentation are made with great care and the manuscript is very well written. It was a joy to read it! I thus recommend the manuscript for publication with minor revision.

Minor comments: p.1 line 19: "According to the kappa-Köhler simulations. . ..." It would be good to indicate if the simulation includes possible feedbacks on the supersaturation due to increased number of CCN at a given supersaturation or if the supersaturation is assumed constant. I would recommend that this is clarified also in other sections.

p.2 line 1: If "certain meteorological conditions" do not refer directly to peak supersaturation, but rather factors that governs the production of condensable water, like updraft, the ability of particles to activate to cloud drops also depends on the aerosol particle number concentration (influencing the critical supersaturation).

p.2 line 20: "spectrum" should be "spectra".

p.7 line 27: Here the term "cloud-free" is used and the same term is used in the figure text for figure 1, but in the legend inside the figure the term "clear" is used. I would recommend that they are made consistent.

p.8 line 31 and top of page 9: It says that event #2 (with polluted air from north east) is characterized by small fraction less-hygroscopic particles and event #1 (with winds from west) has a high fraction less-hygroscopic particles (did I get it right?). Isn't this the opposite compared to the general picture given in the figures S2-S4?

p.10 line 14-17: I do not know how to interpreat the sentence starting with "In our case,...". Does it mean that a linear regression fits the data better than the power law fit? If so, could there be any reasons for that? For example that it is mainly the number fraction of less-hygroscopic particles that gives the variation in kappa? Related to this: in figure 2a a $R^2$ value of -1.99 is given. Do you consider this real?

p.10 line 21: Can you quantify "considerable uncertainties"?

Section 3.5: Specify if the supersaturation is assumed constant or if feedbacks related to the water vapour sink from activated droplets is included.

Supplement: Be consistent in the naming of sectors between figure 1 (S1-S5) and table S2 (1-5). Maybe something like WS1-5 could be used, not to confuse with figure labels in the supplement?

---

## Referee Comment (RC3) · M. Gysel (Referee) · 7 Jun 2016

**Review of the manuscript entitled „In-cloud measurements highlight the role of aerosol hygroscopicity in cloud droplet formation " from Olli Väisänen et al.**

This study investigates the role of aerosol hygroscopicity on its ability to form cloud droplets. Many previous studies have addressed the roles played by particle size and particle hygroscopicity on the ability of such particles to act as cloud condensation nuclei (CCN). However, this was for the most part done based on theoretical considerations, laboratory studies and/or simulated droplet formation on ambient aerosol in CCN counters. This study is one of very few studies that directly investigated the activation of aerosol into the droplets of atmospheric clouds as a function of their size and hygroscopicity. State-of-the-art experimental methods were applied to independently determine the hygroscopicity distributions of the total aerosol and the interstitial aerosol, based on which they could clearly show that the less hygroscopic particles are much less efficient in forming cloud droplets compared to the more hygroscopic particles of equal size. While this essentially confirms expectations, it is still a very valuable result as it directly affects how local/regional emissions contribute to cloud droplet number and to what extent the less hygroscopic particles can be processed by clouds.

The manuscript is generally well written, concise and within the scope of ACP. The data analysis approaches seem appropriate for the most part. However, in two cases I am not sure whether the results are internally consistent. Besides, one figure, which is essentially just an internal consistency check of the data analysis approaches, comes as if it was an independent result (see major comments). The "minor comments" are largely just meant to clarify e few things, to improve the notation in equations in order to avoid ambiguities and to provide several ideas for additional analyses. I do not expect that the latter should all be addressed in a comprehensive manner (nor would I expect exhaustive rebuttals for most of them).

In conclusion, I recommend this interesting and relevant manuscript for publication in ACP after the most relevant comments have been addressed by the authors.

**Major comments:**

1) P.8, l.21-22:
"Typically, the activation efficiencies calculated by Eq. (9) ($f_{act,GF>0.80}$) appeared somewhat larger than the DMPS derived values ($f_{act,DMPS}$)." – I claim that this is impossible! It must be caused by errors in the calculations (e.g. from inconsistent choice of GF-PDF normalization approach and integration of the GF-PDFs in Equation 9 or from comparing at different times). Let me explain: "GF>0.80" means that you integrate over the whole GF-range of the GF-PDF. The fact that the GF-PDF, as used in Equation 9, must be normalized to unit area, directly implies that, for "GF>0.80", Equation 9 simplifies to
$f_{act,GF>0.80}(D_p)= ( dN_{tot}(D_p)/dlogD_p – dN_{int}(D_p)/dlogD_p ) / dN_{tot}(D_p)/dlogD_p$
and this is nothing else than how I would define $f_{act,DMPS}(D_p)$. Or in other words: there is no more HTDMA derived information left in $f_{act,GF>0.80}(D_p)$, only DMPS-derived information. Therefore the poor time resolution or other potential issues with the HTDMA measurement cannot influence the result.
Please reconsider your calculations and add the results from all other available measurements to Table 1 too, if there was really a mistake in the previous approach.

2) Table 1:
Shouldn't the following equality hold for the data shown in Table 1?
$$f_{act,GF>0.80}(D_p) = f_{GF<1.25}(D_p) * f_{act,GF<1.25}(D_p) + (1 - f_{GF<1.25}(D_p)) * f_{act,GF>1.25}(D_p)$$
(simply use Equation 9 to get this). It does seem to be fulfilled for some columns of Table 1 but not for others. Please check your data or clarify the manuscript if I misinterpreted the meaning of Equation 9 or $f_{act}$ or so.

3) Figure 3 and first paragraph of Section 3.4:
The comparison of cloud droplet predictions made with considering the hygroscopic mixing state or with assuming internal mixture (and using $\kappa_{avg}$) is useful, as it confirms previous results on the sensitivity of CCN predictions to simplified treatment of mixing state for the effect of mixing state of total cloud droplet concentration in real clouds.
By contrast, the comparison of HTDMA+DMPS$_{tot}$+S$_{c,eff}$-predictions of cloud droplet number concentration with those derived from the DMPS$_{tot}$ minus DMPS$_{int}$ measurements is meaningless because it is circular argumentation! Equation 8 was first used to infer the S$_{c,eff}$ from the HTDMA, DMPS$_{tot}$ and DMPS$_{int}$ measurements. The S$_{c,eff}$ obtained in this manner was then used, along with the HTDMA + DMPS$_{tot}$ data, for cloud droplet prediction to be compared with those derived from DMPS$_{tot}$ minus DMPS$_{int}$ measurements as shown in Figure 3. Consequently, this analysis has nothing to do with a closure study, instead it is simply an internal consistency check of the approach to infer S$_{c,eff}$ and to predict cloud droplet number. Therefore, this part of the figure (i.e. the red data points) has to be removed and the discussion needs to be adapted.
Nevertheless, it is worth to have a closer look into Figure 3. The fact that the grey points have higher correlation and slope closer to unity than the red points is unexpected. If the cloud droplet prediction is so insensitive to assuming internal mixing state, why is then the "internal consistency check" of inferring and re-applying the effective supersaturation from/to the HTDMA measurements so poor (referring to slope and particularly the scatter of the red points)? Is this related to improper treatment of activation plateau values that differ from unity (see separate comment below concerning this potential issue)? Or is the minimization approach applied to infer the effective supersaturation (Equation 8) not suitable? Other reasons?

**Minor comments:**

4) Abstract:
Well written but for one missing element: The range of supersaturations occurring in the  is not mentioned. This would be worthwhile as they appear to be rather low, which increases the sensitivity of CCN number concentration to hygroscopicity/composition compared to high supersaturation.

5) P.3, l.26:
It might be worthwhile to mention how the visibility was measured.

6) Sect. 2.3:
A PM$_{1.0}$ impactor was used in the interstitial inlet to remove the activated cloud droplets just leaving the interstitial aerosol behind. However, the mode of none-activated droplets in stable equilibrium within the cloud may potentially extend to diameters larger than 1 µm, if the peak supersaturation at cloud formation was very low (the lower the peak supersaturation, the larger the activation cut-off diameter, the larger the maximal diameter of non-activated interstitial

particles). In such a case, the largest and most hygroscopic particles of the true interstitial aerosol in the cloud would be missing in the aerosol sample measured behind the interstitial inlet. Hammer et al. (2014) address this issue in more detail. Can you exclude this potential artefact based on supporting data such as droplet size distributions or estimates of the maximal possible equilibrium droplet size for the clouds formed with the lowest peak supersaturation? Hammer, E., Gysel, M., Roberts, G. C., Elias, T., Hofer, J., Hoyle, C. R., Bukowiecki, N., Dupont, J.-C., Burnet, F., Baltensperger, U., and Weingartner, E.: Size-dependent particle activation properties in fog during the ParisFog 2012/13 field campaign. *Atmos. Chem. Phys.*, **14**, 10517-10533, doi:10.5194/acp-14-10517-2014, 2014.

7) Sect. 2.4:
The residence time between humidifier and DMA is provided (2s). The residence time within the humidifier might also be of interest, possibly even including a brief remark on the reasons for choosing a rather short or rather long residence time.

8) Equation 2 and other instances in Sect. 2.5 and possibly the rest of the manuscript:
This equation (Köhler-equation) describes the equilibrium water vapour saturation ratio ($S_{eq}$) as a function of particle hygroscopicity, diameter, etc. By contrast, the critical saturation ratio ($S_c$) is the maximum of the Köhler curve described with Equation 2. You are using the same symbol for these two different quantities. This needs to be fixed.

9) Line just below Equation 3, "…where $V_s$ and $V_w$ are the soluble and water volumes…":
In general, $V_s$ is the volume of the whole dry particle, not just the soluble components. Insoluble fractions, if present, are accounted for with the κ-value.

10) Equation 5: This analytical solution for the critical saturation ratio contains mathematical approximations which become increasingly inaccurate with decreasing critical saturation ratio. Therefore, this equation should only be used for qualitative purposes, whereas a numerical solution of the Köhler equation must be implemented for quantitative purposes.

11) Equation 5 and line just below:
This is now the critical saturation ratio. It should be explained what it is – in contrast to the equilibrium saturation ratio appearing in Equation 2 – and the symbol should be defined.

12) Equations 6 and 7:
It could be mentioned that the approach and equations used to predict the CCN number concentration from particle number size distribution (from SMPS) and GF-PDFs (from HTDMA) is identical to the approach introduced by Kammermann et al., 2010b (cf. their Equations 2&3).

13) P.6, l. 5-8:
This step also involves interpolation in time (besides inter-/extrapolation in size).

14) P.6, l. 11ff:
"Nonetheless, it has to be noted that above 200 nm, hygroscopicity is quite rarely a limiting factor and the most crucial activation characteristics are dependent on particle properties between the ~80 nm and ~200 nm sizes." – Isn't this statement somewhat in conflict with your result that a substantial portion of the non-hygroscopic particles remains interstitial (at the largest diameter covered by your measurements)? Therefore you need the additional argument

that the number fraction of non-hygroscopic particles depends only weakly on size across the size range relevant in this context.

15) P.6, l.13ff:
"Secondly, the method assumes that the subsaturated hygroscopicities are representative for supersaturated conditions. Such an assumption is not always totally valid and discrepancies between the two saturation regimes have been reported based on laboratory and field experiments…" – The study by Jurányi et al. (2013) could also be referenced here, as one of the examples that found very good closure between sub- and supersaturated regimes for externally mixed urban aerosol.

16) P6., l.19ff:
Comments on the approach to estimate the effective peak supersaturation (PS: I'd suggest $S_{c,eff}$ rather than $S_{eff}$ as symbol):

a. I suggest to start with a brief explanation of the concept behind estimating $S_{c,eff}$, possibly also referring to Hammer et al. (2014). This would be likely be helpful for the "average" reader of this manuscript.

b. If entrainment occurs or in the case of partially/fully glaciated clouds the plateau value of $f_{act,DMPS}$, i.e. the value $f_{act,DMPS}$ takes at large diameters at which also the non-hygroscopic particle activate, may be substantially smaller than unity (e.g. Fig. 3 in Verheggen et al., 2007). The minimization approach given in Equation 8 would cause a bias for such a scenario. – Did you observe evidence for entrainment and/or glaciation or did the plateau value of $f_{act,DMPS}$ always reach unity?
Verheggen, B., Cozic, J., Weingartner, E., Bower, K., Mertes, S., Connolly, P., Gallagher, M., Flynn, M., Choularton, T., and Baltensperger, U.: Aerosol partitioning between the interstitial and the condensed phase in mixed-phase clouds. *J. Geophys. Res.*, **112**, D23202, doi:10.1029/2007JD008714, 2007.

c. Our experience from similar measurements at the Jungfraujoch research station is that the diameter range across which $f_{act,DMPS}$ increases from 0 to 1 is much broader than can be explained with the heterogeneity of the aerosol in terms of mixing state/GF-PDF (the JFJ-aerosol is rather internally mixed). This indicates that the width of $f_{act,DMPS}$ is mainly driven by heterogeneity of $S_{c,eff}$ on small spatial scales due to e.g. turbulence. What does it look like in your case (the aerosol at Puijo is obviously much more externally mixed than that observed at the Jungfraujoch)? Can the shape of $f_{act,DMPS}(D_p)$ be explained with the external mixing alone? This question possibly goes beyond the main focus of this paper, but it might still be worth looking at it. You have the data at hand and could possibly produce a supplementary figure using representative examples).

d. An alternative approach would be to fit something like a sigmoid curve into $f_{act,DMPS}(D_p)$ to obtain an effective activation cut-off diameter (half rise). Inserting this cut-off diameter and $\kappa_{avg}(D_p)$ into Köhler theory then provides $S_{c,eff}$ under the assumption of internal mixing. I would expect that these values are very similar to those obtained with your approach accounting for external mixture. If not, you should comment on the fact

that proper treatment of mixing state is crucial for inferring $S_{c,eff}$ when dealing with clouds formed on externally mixed aerosol.

   e.   Did you define $f_{act,DMPS}(D_p)$?

17) P.6, l.26ff:
First, this paragraph belongs above the paragraph describing how you infer $S_{c,eff}$, as this is still about predicting CCN number concentration from HTDMA data, if I got that right. Second, you may have to explain how you obtain $\kappa_{avg}(D_p)$.

18) Sect. 3.1:
Some at least partially glaciated clouds wouldn't be surprising if ambient temperature was sometimes below zero (minimum was -9.7 °C).

19) Figure 1, bottom row:
GF-PDFs shown here are normalized to unit area. I wonder whether it would be instructive to add an extra row (or replace the current bottom row) with a version in which you re-normalize the GF-PDFs as follows. For the total aerosol, multiply the GF-PDF that is already normalized to unit area with $dN_{tot}/dlogD_p(D_p)$. And equivalently for the interstitial aerosol. The area between the curves representing the total and interstitial GF-PDFs for equal size would then directly correspond to the activated particles. Furthermore, normalized in this manner, the bottom row of Figure 1 would then be are more close graphical representation of what you calculate with Equation 9. You could even add an extra row of panels that shows $f_{act,GF}$ for each diameter and GF-resolved, i.e. as a function of GF rather than integrated over a GF range.

20) P.7, l.20:
Was the shift of the more hygroscopic mode towards larger GF with increasing particles size less or more than what can be explained by the size dependence of the GF imposed by the Kelvin effect?

21) P.7, l21ff:
The observations by Laborde et al. (2013) in Paris revealed even a little more detail. There was very clear evidence that particles from fresh traffic emissions appeared mainly at GF≈1.0, whereas particles from wood burning appeared mainly at GF≈1.1, together forming the "non-hygroscopic" mode in the HTDMA. I have added this detailed remark because the GF-PDFs shown in the bottom row of your Figure 1 for particles with diameters of 120 nm and 150 nm seem to provide evidence that the cloud droplet active fraction differs slightly between GF≈1.0 and GF≈1.1. Could you confirm this or is this difference within uncertainty?

22) Equation 9:
$N_{tot}(D_p)$ should be replaced by $dN_{tot}(D_p)/dlogD_p$, shouldn't it? And so for $N_{int}(D_p)$? – You define $N_{tot}$ as: "…where $N_{tot}$ and $N_{int}$ are the total and interstitial number concentrations…". This rather rather sounds as if $N_{tot}(D_p)$ was representing a cumulative number concentration, which would be wrong in Equation 9 (as I understand the purpose of Equation 9).
PS: you could of course also choose $dN_{tot}(D_p)/dD_p$ instead of $dN_{tot}(D_p)/dlogD_p$ as the factor in between those two eventually cancels out.
Besides: I recommend adding another line to Equation 9, in which you rearrange it as follows:

$f_{act,GF1<GF<GF2}(D_p, GF_1, GF_2) = ( dN_{tot}(D_p)/dlogD_p * f_{tot,GF1<GF<GF2}(D_p) - dN_{int}(D_p)/dlogD_p * f_{int,GF1<GF<GF2}(D_p) ) / dN_{tot}(D_p)/dlogD_p * f_{tot,GF1<GF<GF2}(D_p)$

where $f_{tot,GF1<GF<GF2}(D_p)$ is the number fraction of particles (total aerosol) with dry diameter $D_p$ and GF between GF1 and GF2 (and equivalent for the interstitial particles). This addition should help in understanding the meaning of Equation 9.

23) P.9, l.3ff:
"The most interesting remark concerns the difference between the low and high hygroscopicity particles at 120 and 150 nm. While the activation efficiency of total aerosol and more hygroscopic particles increases with size, the less hygroscopic particle mode remains almost non-activated." – The "size dependence" mentioned in this statement is distracting from the main message. In my view Table 1 already captures the central and very nice results of your study, which is: "...., the cloud droplet activated fraction of the less hygroscopic particles is much smaller than that of the more hygroscopic particles of equal size...which confirms that cloud droplet activation critically depends on particle hygroscopicity for particle sizes for all sizes in the range of the droplet activation cut-off...." The very nice thing is that you showed this, which is expected based theory and hygroscopicity-resolved HTDMA-CCN closure studies, for the activation of atmospheric aerosols in atmospheric clouds. Personally I would focus on this, i.e. comparing less versus more hygroscopic at equal size, and address size dependence in the next paragraph.

24) P.9, l.11ff:
"Here, the residual aerosol-properties were estimated indirectly by using the hourly averaged total and interstitial GF-PDFs and their actual number concentrations." – Which factors did you apply to the normalized GF-PDFs, $dN_{tot}(D_p)/dlogD_p$ and $dN_{int}(D_p)/dlogD_p$ for the total and interstitial inlets or did you revert the normalization factor of the GF-PDFs with the normalization factor that had been applied? I would believe that the former is the better choice, if the total and interstitial DMPS measurements are corrected such that they are identical for out of cloud measurements. However, this is just a subtlety.
PS: Applying the number of counts of the HTDMA raw measurements would be "wrong" because the detection probability in the HTDMA is GF-dependent for a fixed dry size. However, again just a small but still systematic bias.

25) P.9, l.10-17: why do you not discuss the size dependence of the activated fractions of the more and of the less hygroscopic particles in this paragraph? There are good reasons for how they depend on size and why there is hardly any difference between total and interstitial inlet at the smallest covered size.

26) Figure 1:
According the legend in the bottom row of Figure 1 the difference of the average GF between total and interstitial inlet is 0.04 for the two dry diameters 80 nm and 120 nm. However, the difference between total and interstitial seems to be much larger for 120 nm compared to 80 nm when looking at the bottom row of Figure 1. Please check carefully and adapt the figure and discussion on P.9 l.10-17 if needed.

27) P.9, l.18 ff:

"To our knowledge, this is one of the very few studies characterizing the hygroscopic properties of different in-cloud aerosol populations." – there might exist some CCN based literature on this topic; you could check for authors like U. Pöschl and D. Rose.

Concerning chemical composition: you could check for SP2-based studies by J. Schroder et al. This might potentially link in to the behaviour of the less-hygroscopic particles.

28) Figure 2 and associated discussion:

Relevant analysis, however, somewhat incomplete. Equation 5 tells us that the three parameters $S_c$, $\kappa$ and $D_{50}$ are related to each other. Therefore, the relation between the three of them should be reflected in the analysis, figures and discussion. Some thoughts on this:

a. The dependence of $D_{50}$ on $\kappa$ could be the result of cross-correlation rather than causality. You should confirm that $\kappa$ and $S_c$ are not correlated to make your result stronger. This is definitely required before you make the statement at the end of Section 3.3.

b. Color code: the most relevant information I seem to learn from the colour code is that the variability of $\kappa$ is for the most part driven by the variability of the number fraction of less and more hygroscopic particles rather than the variability of the respective mean GFs of these two modes. Correct? This would be better seen from a scatter plot of $\kappa$ versus $f_{GF<1.25}$.

c. Fit curve: the fit curve can be quantitatively interpreted, i.e. it provides you an estimate about the $S_{c,eff}$ "averaged" over the whole data set. Is this value consistent with your other analyses of $S_{c,eff}$? Caveat: Equation 5 is an approximation, which is likely not accurate for the rather low critical supersaturations you are dealing with. ➔ see next comment.

d. I suggest you include multiple theoretical lines in Figure 2 that show $D_{50}$ versus $\kappa$ for different $S_c$ (based on unbiased numerical solutions rather than the approximate Equation 5). These theoretical lines might possibly save you the trouble of including a fit curve. Additionally you should choose the $S_{c,eff}$ as colour scale for the data points (you can have multiple versions with different colour scales if you like to keep your old colour scale too). This will give you a more complete picture on the influence of $\kappa$, $S_{c,eff}$ and also "measurement noise" on the variability of $D_{50}$.

e. You should also create a figure in which you swap the roles of $\kappa$ and $S_{c,eff}$, i.e. you plot $D_{50}$ vs $S_{c,eff}$ and choose $\kappa_{avg}$ (I'd say for 120 nm or 150 nm or a value interpolated to the mean D50) as colour scale (theoretical lines should also be added). How does it compare with the figure suggested above?

f. The outlier in Figure 2: is it an outlier in the sense of "cannot be explained" or do you have independent evidence that the very high $D_{50}$ could possibly be caused by exceptionally low supersaturation (you cannot use $S_{c,eff}$. to argue as $S_{c,eff}$ is inferred from D50)?

29) P.10, l.28-29:
"…the estimated peak supersaturations… …they provide some valuable information about the in-cloud conditions…". – The droplet activation happens at the initial stages of cloud formation.

30) Last paragraph of Section 3.4 (comparison of supersaturations with literature):
Hammer et al. (2014) reported a systematic difference in observed peak supersaturations for the two prevalent wind directions, which could be explained by differences in the orographic forcing (steep vs gentle mountain slopes). What are the cloud formation mechanisms for the clouds probed at Puijo (and Pallas)? Are the lower peak supersaturations at those two sites possibly related to weaker orographic forcing compared to the Puy de Dôme and Jungfraujoch sites?

31) Figure 4 and associated discussion:
The susceptibility of cloud droplet concentration to hygroscopicity can be quite asymmetric with respect to increase vs decrease of $\kappa$ (see e.g. Figure 8 in Juranyi et al., 2010, or other studies that did similar sensitivity analyses for CCN number concentrations). Instead of just considering the case "no less hygroscopic particles at all" (➔ higher $\kappa$), you could additionally consider the case "no more hygroscopic particles at all" (➔ lower $\kappa$) for the sensitivity analysis presented in your Figure 4 and Table 2.
Jurányi, Z., Gysel, M., Weingartner, E., DeCarlo, P. F., Kammermann, L., and Baltensperger, U.: Measured and modelled cloud condensation nuclei number concentration at the high alpine site Jungfraujoch. *Atmos. Chem. Phys.*, **10**, 7891-7906, doi:10.5194/acp-10-7891-2010, 2010.
PS: further down in the manuscript it became clear why you specifically look at positive deviations. You could try to clarify this earlier.

32) Concerning difference of the activation behaviour of the two modes:
As the aerosol at Puijo appears to have two rather well separated hygroscopicity modes, and since you prove that this directly affects the cloud droplet formation ability, you could quantify the expected difference of activation cut-off diameter for these two modes, if you like. One option would be the following: from "every" HTDMA measurement you can infer $S_{c,eff}$, $\kappa_{avg,GF<1.25}$ and $\kappa_{avg,GF>1.25}$. This allows to infer $D_{50,GF<1.25}$ and $D_{50,GF>1.25}$. Plotting $D_{50,GF<1.25}$ and $D_{50,GF>1.25}$ versus $S_{c,eff}$ then gives a fair idea of the activation cut-off diameter of the two modes, which is for example relevant for the threshold size down to which the particles in either mode can undergo cloud processing under the conditions in clouds at Puijo. (The only thing you would have to think about is how to deal with the diameter depends of hygroscopicity.)

33) There is another potentially interesting question you could look at if you like: while assuming an internally mixed aerosol can provide very good estimates of the total CCN number concentration, if properly done, it will not give an accurate answer concerning the respective contributions of the background aerosol and local/regional emissions to CCN number (with the picture in mind that the less hygroscopic mode is of local/regional origin). Based on your data set you could make at least a crude estimate of how the number fraction of local/regional particles compares between total aerosol and those particles that formed cloud droplets (pulling the idea of the previous comment even a little further). Or in other words: your data set seems to imply that most particles of local/regional origin have to undergo quite some atmospheric aging processes until they start participating in cloud droplet formation, doesn't it?

34) P.11, l.26-27:

"Understandably, by suppressing the size-dependent variations in chemical composition, the activation curves become steeper and the D50s decrease." – This is unclear. You only present data from a single size, so how can size dependence be supressed? To my understanding the D50s decrease because you make the particles more hygroscopic! Please clarify how you mean this.

Technical corrections:

35) P.7, l24:

In the context of HTDMA measurements I would speak of "non-hygroscopic" particles for GF=1.0 rather than hydrophobic. I'd rather use the latter term in the context of measurements that are sensitive to "wettability", i.e. adsorption or contact angle or similar.

---

## Author Comment (AC1) · 13 Jul 2016

**Authors' response to Anonymous Referee #1**

We would like to thank the Anonymous referee #1 for his/her valuable comments and suggestions. All the comments have been taken into account and the manuscript has been revised accordingly. Below, the reviewer's comments are written in italics and followed by authors' responses.

**General comments**

*The manuscript "In-cloud measurements highlight the role of aerosol hygroscopicity in cloud droplet formation" investigated the hygroscopic growth distribution of particles at three sizes in cloud events and the hygroscopicity-dependent droplet activation at a tower station in Puijo, Finland. The authors found the bimodal hygroscopic growth factor distribution. This study further examined the changes of droplet activation properties if the less hygroscopic particle fractions have the same hygroscopicity as the higher hygroscopic fractions. By this, the manuscript highlighted the important role of chemical composition, namely hygroscopicity, in cloud droplet activation. This manuscript is generally well written and provides an evaluable ca se study on the role of chemical composition in cloud droplet activation. Together with other previous studies, it helps understand the role of size and chemical composition in CCN activation in a balanced way. Realizing that both size and chemical composition are important in CCN activation rather than over-stressing one over the other benefit the CCN studies. But before it is published on ACP, the manuscript need address the following questions.*

1. *Some important details need further clarification. For example, in Pg. 4, lines 14. The authors stated that "full data inversion was applied..., including the corrections for sampling losses, multiple charge probabilities, instrument transfer function and particle count efficiencies". The corrections are important for the data evaluation, but it is not clear for readers. These details should be provided in the Supplement or at minimum be referred to specific references in a detailed way. Another example is that how the D50 in Fig. 2 and D50 in Fig. 4 were exactly derived is missing. And "$f_{act,DMPS}(D_p)$" in Eq. 8 was not defined. More examples are included in the "specific comments" part.*

   **Response:** Regarding the reviewer's comment on DMPS inversion and corrections, we have added a reference to Wiedensohler et al. (2012) summarizing the requirements for DMPS data evaluation and inversion in detail. In addition, the D50 values were obtained from the DMPS and HTDMA derived activation curves via linear interpolation. This is now clarified in the text.

2. *The authors attributed the "less hygroscopic" particles to be "originating from local anthropogenic sources". In the conclusion, it was stated that "our result clearly demonstrates the importance of correct treatment of anthropogenic aerosols, their hygroscopicity and the effect of atmospheric aging, when estimating the CCN concentration." I guess the authors specially meant the anthropogenic organic aerosol and carbonaceous aerosol since the $(NH_4)_2SO_4$ or $NH_4NO_3$ are highly hygroscopic. While I agree it is important to correctly treat aerosols with both low and high hygroscopicity, it should also be noted that the biogenic secondary organic aerosols can also have a low kappa (e.g around 0.1 as in Fig.2). They could also contribute to the total particles even when the winds were from "anthropogenic" sectors since air masses could pass over the forest regions before reaching the measurement site (a backward trajectory may help clarify whether this could be true). Moreover, without detailed aerosol chemical composition for source apportion, it is only a plausible speculation here that the aerosol is "originating from local anthropogenic emission. I suggest the authors to use more precise wording here in the conclusion and in the abstract.*

**Response:** With the sentence "*our result clearly demonstrates the importance of correct treatment of anthropogenic aerosols, their hygroscopicity and the effect of atmospheric aging, when estimating the CCN concentration*" we refer to anthropogenic organics. We have clarified this in the corrected manuscript. As pointed out by the referee we can't exclude the contribution of biogenic organics in non-activated fraction. Nonetheless, according to a wind sector analysis (see figures S2-S4), the two highway sectors were characterized by higher $f_{GF<1.25}$ than the clean sectors indicating clear contribution of traffic on the increased fraction of less hygroscopic particles. We have modified the text both in the abstract and in the conclusion section accordingly.

**Specific comments**

1.  **Comment:** *Pg. 1, line 13, as mentioned as above, "reflecting the varying presence of fresh anthropogenic particle emission" is based on plausible speculation. Some hedge words are recommended here.*

    **Response:** Text modified.

2.  **Comment:** *Pg. 4, line 22, please define the "Vienna type DMA".*

    **Response:** We have added a reference to Winklmayr et al. (1991) describing the basic characteristics of Vienna type DMAs.

3.  **Comment:** *Sect. 2.4, What is the residence time of the particles in the humidifier?*

    **Response:** The residence time inside the humidifier was approximately 0.2 seconds, after which the particles spent ~2 s in elevated humidity before being size selected by DMA2. This piece of information is now added to the manuscript.

4.  **Comment:** *Pg. 6, line 22 Eq. 8, as mentioned above, "$f_{act,DMPS}(Dp)$" was not defined.*

    **Response:** The DMPS derived activation efficiency, $f_{act,DMPS}(D_p)$, is now defined both textually and with an equation.

5.  **Comment:** *Pg. 8, line 1, "...during the changing inlet period...", if I understand correctly, the inlet alternated every 6 min between the total aerosol and interstitial aerosol measurement. By using "changing inlet period", did the authors suggest there were some periods when inlet was not changing? Please clarify.*

    **Response:** Yes, the DMPS was connected to the twin inlet system with an automated valve that was switching between the two sample lines in 6-minute intervals. This setup was untouched thorough the campaign.

    In contrast, the HTDMA was attached to the twin inlet system only for a 4-week sub-period through an external valve system (different from the one operating the DMPS). For the sake of clarity, this period is now defined as a "twin inlet period" and referred as such thorough the paper. During the campaign days prior to and after the twin inlet period, the HTDMA was sampling only from the total line.

6. **Comment:** *Pg. 9 lines 4-5, "... the less hygroscopic particle mode remains almost non-activated" is not fully convincing to me. From event #1 and # 3, some fractions (12-33%) of the particles still activated. What about other clouds events in the 15 events? The activated fractions are typically around 0 or as in event #2 or more like event #1 and #3? Maybe re-phase this sentence, saying something like "the activated fraction is much lower (value x-value y%)".*

**Response:** In total, nine cloud events were observed during the twin inlet period. Three of them were included in the original manuscript and one of them was characterized by negligible less hygroscopic fraction. For the five remaining cloud events, the activated fractions (GF < 1.25) were 21-20-0-19-0 (%) and 27-25-0-17-0 (%) at 120 and 150 nm, respectively. Although these values are very similar to those reported in the original manuscript, their reliability can't be fully confirmed due to low data coverage and/or disagreement between the HTDMA and DMPS derived total activated fractions.

After revising the calculation method based on comments by Dr. Martin Gysel, we have now presenting the data for four different cloud events. This part of the text is re-written keeping in mind the reviewer's comments above.

7. **Comment:** *A related question to the last question. In Fig. 1, what are the unit and value of y-axis in these graphs? I suppose the value is the number concentration of particle with given GF value. If I understand correctly, in the bottle middle panel of Fig.1, the residue concentration divided by total concentration would yield the activated fraction as a function of GF. Integrating the residue concentration for the range of GF<1.25 divided by the integrated total concentration would yield the activated fraction of particle with GF<1.25 ($f_{act,GF<1.25}$). It seems that the values derived in this way would not be close to zero for either 120 nm particles or 150 nm particles. Could the authors explain this?*

**Response:** In Fig. 1, the GF distributions are plotted as probability density functions. This means that the initial GF distributions are normalized according to their actual particle number concentrations. Thus, their integrated surface areas are always equal to one. As a result, the y-axis is actually unitless, and estimating the activated fractions would require additional information about the total and interstitial particle concentrations.

8. **Comment:** *Pg. 9, line 32, "... in line with the hygroscopic discrepancies...". It is not clear for me what the authors meant by "discrepancies". Do they mean the different activated fraction between the less and more hygroscopic particles? If so, please phrase it precisely.*

**Response:** We agree that this statement wasn't perfectly clear. It is now rephrased as "in line with our observations on the distinctive hygroscopicity of activated and non-activated particles"

9. **Comment:** *Pg. 10, lines 15-17, "...hygroscopicity of accumulation mode particles explained up to 57-58% of the observed variance in D50,...especially, the varying supersaturation"., from Fig. 2, the middle and bottle panel, most data points are converged to be close to the line of D50=Ak-1/3. Does this indicate that the variability of the supersaturation is quite small? And do these graphs include all the data from 15 clouds events?*

**Response:** As we report later in the paper, the estimated supersaturations ranged from 0.16 to 0.29%. We agree that this range of variation is reasonably narrow compared to global observations, and larger fluctuations would definitely lead to increased deviation.

Figure 2 includes data from all the observed cloud hours. This is now clarified in the text.

10. **Comment:** *Pg. 10, lines 20-21, the authors stated that "...the data points with considerable uncertainties were omitted...". Could the authors precisely describe what data were classified as ones "with considerable uncertainties" since how the data were omit would affect the correlation in Fig. 3. Also does the "cloud droplet nuclei spectra" refer to the spectra derived from DMPS? Please clarify.*

**Response:** Regarding the omitted cases, the following sentences were added:

"Reproducibility of the DMPS derived activation curves and cloud droplet nuclei spectra was confirmed visually, and the data points with considerable differences were omitted from the following analyses. In most of these cases, the estimated activation curves appeared clearly steeper than the measured ones, leading to both an underestimation of activated fraction at $D_p$ < D50 and an overestimation of activated fraction at $D_p$ > D50. Alternatively, some of the estimated curves managed to reproduce the correct behavior at smaller sizes ($D_p$ < 150 nm) but overestimated the activation at the size range of extrapolated GF-PDFs. Although these uncertainties caused only a small net bias to droplet estimations, they indicate that the estimated GF surfaces may have failed to replicate the real ambient conditions."

As mentioned above, these uncertainties didn't really increase the bias in droplet predictions (see Fig. 1 attached below). However, they indicate that the linear interpolation between the measured GF-PDFs and furthermore, extrapolation towards larger sizes, may have failed to predict the size-dependence adequately. Therefore, they must be removed from the data set to minimize the uncertainties in the following analyses (Sect. 3.5 in the manuscript).

[Figure]

**Fig. 1**

11. **Comment:** *Pg. 13, line 4, as mentioned above, "the less hygroscopic particles originating from local anthropogenic emissions" is only a speculation. I suggest to add a hedge word here, e.g. likely.*

**Response:** Text modified.

**Technical comments**

1. **Comment:** *Pg 4, line 11, 'The instrument was..." should be "instruments were".*

    **Response:** Corrected as suggested.

2. **Comment:** *Pg 5, line 1, "...using the TDMAinversion..."", a space is missing after TDMA.*

    **Response:** This is probably just an accidental misunderstanding. We are actually writing: "…by using the TDMAinv inversion toolkit".

3. **Comment:** *Pg. 10, line 2, "... the above results..." should be "...the results above...".*

    **Response:** Rephrased as: "According to our observations…"

4. **Comment:** *Pg. 12, line 1, "... the absolute change in hygroscopicity increased...", is "hygroscopicity" the right word? Or growth factor?*

    **Response:** Corrected as suggested.

5. **Comment:** *Pg. 12, line 12, "Similarly to our measurement..." should be "similar".*

    **Response:** Rephrased as: "Analogous to our observations…"

[revised manuscript text omitted]

The difference between the activated and non-activated particles is also illustrated in Fig. 1 (lower panel) where the average GFs and GF-PDFs are presented separately for total, interstitial and residual aerosol populations. Here, the residual aerosol properties were estimated indirectly by using the hourly averaged total and interstitial GF-PDFs and the respective ambient particle concentrations as follows:

$$c_{res}\left(GF, D_p\right) = \frac{dN_{tot}}{d\log D_p}\left(D_p\right) \times \text{GF-PDF}_{tot}\left(GF, D_p\right) - \frac{dN_{int}}{d\log D_p}\left(D_p\right) \times \text{GF-PDF}_{int}\left(GF, D_p\right). \tag{11}$$

Apparently, the relative contribution of less hygroscopic particles is the strongest in the interstitial population and the more hygroscopic mode appears distinctly only in the total and estimated residual aerosol. This is also reflected by the average growth factors. At $D_p$ = 120  and 150 nm, the GFs of cloud droplet residuals are approximately 18% higher than those of interstitial particles. In terms of $\kappa$ values, this discrepancy would correspond to a difference up to 65%–

75% depending on particle size. Again, it should be remarked that similar to activated fractions, the residual GF distributions ($c_{res}$) can appear locally negative if the interstitial concentrations are higher than the respective total concentrations. Before converting these distributions into GF-PDFs shown in Fig. 1, the negative values were set to zeros which eventually strengthened the positive peaks appearing in normalized distributions. As a result, for example the less hygroscopic particle mode appearing in the estimated residual aerosol can be partially attributed to methodological uncertainties.

To our knowledge, this is one of the very few studies characterizing the hygroscopic properties of different in-cloud aerosol populations. Previously, Svenningsson et al. (1994) studied the aerosol hygroscopicity and its relationship to cloud droplet activation at Kleiner Feldberg in Germany. Interstitial aerosol hygroscopicity was measured during cloud events, and by assuming that the air mass was the same, it was compared to the total aerosol sampled during the following clear sky conditions. The less hygroscopic particle fraction was substantially higher in the interstitial population, indicating that the more hygroscopic particles were scavenged into the cloud droplets more efficiently than the less hygroscopic ones. This observation was confirmed in a case study, where a counter flow virtual impactor (CVI) was used to separate the cloud droplets from the total aerosol, so that the hygroscopicity of cloud droplet residuals could be measured independently. Likewise, Rose et al. (2013) observed a decreasing fraction of available CCN during a precipitative cloud event, suggesting that the more hygroscopic particles were mostly activated into cloud droplets and removed from the air through precipitation. Overall, theour results by Svenningsson et al. (1994)from Puijo are in a good agreement with ourthe observations from Puijo.
[revised manuscript text omitted]

---

## Author Comment (AC2) · 13 Jul 2016

**Authors' response to Anonymous Referee #2**

We'd like to thank the Anonymous referee #2 for his/her positive comments and feedback. Below, the reviewer's comments are written in italics and followed by authors' responses.

*This paper analysis the influence of aerosol particle hygroscopicity on cloud droplet formation, based on field measurements. As pointed out by the authors, there are very few measurements dealing directly with this issue. The issue is at the same time relevant for aerosol-cloud-climate interactions as well as for the transport and lifetimes of air quality relevant pollutants and biogenic compounds in the aerosol. The measurements, data analysis and presentation are made with great care and the manuscript is very well written. It was a joy to read it! I thus recommend the manuscript for publication with minor revision.*

**Minor comments:**

1. **Comment:** *p.1 line 19: "According to the kappa-Köhler simulations....." It would be good to indicate if the simulation includes possible feedbacks on the supersaturation due to increased number of CCN at a given supersaturation or if the supersaturation is assumed constant. I would recommend that this is clarified also in other sections.*

   **Response:** The possible feedback effects on supersaturation were not included in the simulations. This is now clarified in the abstract, Sect. 3.5 and summary.

2. **Comment:** *p.2 line 1: If "certain meteorological conditions" do not refer directly to peak supersaturation, but rather factors that governs the production of condensable water, like updraft, the ability of particles to activate to cloud drops also depends on the aerosol particle number concentration (influencing the critical supersaturation).*

   **Response:** This is a good point. This sentence is now rephrased as: "The particles' ability to activate into cloud droplets at a certain level of supersaturation depends on their size and chemical composition".

3. **Comment:** *p.2 line 20: "spectrum" should be "spectra".*

   **Response:** Corrected as suggested.

4. **Comment:** *p.7 line 27: Here the term "cloud-free" is used and the same term is used in the figure text for figure 1, but in the legend inside the figure the term "clear" is used. I would recommend that they are made consistent.*

   **Response:** Corrected as suggested.

5. **Comment:** *p.8 line 31 and top of page 9: It says that event #2 (with polluted air from north east) is characterized by small fraction less-hygroscopic particles and event #1 (with winds from west) has a high fraction less-hygroscopic particles (did I get it right?). Isn't this the opposite compared to the general picture given in the figures S2-S4?*

   **Response:** Yes, You're correct. According to the supplementary figures, the polluted sector is (on average) characterized by increased fraction of less hygroscopic particles. The three cases presented in

the original manuscript, however, do not precisely follow this trend. This is most likely due to the fact that compared to campaign averages, the temporal scale of cloud events is extremely short and therefore sensitive to natural variations.

6. **Comment:** *p.10 line 14-17: I do not know how to interpret the sentence starting with "In our case,...". Does it mean that a linear regression fits the data better than the power law fit? If so, could there be any reasons for that? For example that it is mainly the number fraction of less-hygroscopic particles that gives the variation in kappa? Related to this: in figure 2a a R2 value of -1.99 is given. Do you consider this real?*

**Response:** Because $R^2$ is not an optimal measure for goodness-of-fit in non-linear regression (e.g. due to the possibility for negative values), we have decided to omit those values from the Fig. 2 (Now, Fig. 3). Due to the fact that the relationship between κ and D50 is not generally linear, the linear regression is applied only for comparison purposes. It is also true that by simply looking at $R^2$ or RMSE values, the linear fit seems to fit the data better than the non-linear one. However, this is most likely just a sheer coincidence resulting from the fact that most of the data points reside in the "flat" part of the non-linear κ-D50 curve.

This paragraph is now rephrased as:
> "…In comparison, applying a linear regression to our data set would result in $R^2$ values of 0.58 (120 nm) and 0.57 (150 nm). Assuming that the relationship between $\kappa$ and D50 could be treated as (locally) linear, these values would indicate that the accumulation mode hygroscopicity explained up to 57–58% of the observed variance in D50, therefore dominating the effect of varying meteorology and especially, the varying supersaturation. Moreover, no correlation was found between the $\kappa$ values and the estimated effective peak supersaturations ($R^2 \sim 0.02$; not shown here)."

7. **Comment:** *p.10 line 21: Can you quantify "considerable uncertainties"?*

**Response:** We have added a few of sentences to clarify, what kind of cases were omitted from the analysis:
> "In most of these cases, the estimated activation curves appeared clearly steeper than the measured ones, leading to both an underestimation of activated fraction at $D_p$ < D50 and an overestimation of activated fraction at $D_p$ > D50. Alternatively, some of the estimated curves managed to reproduce the correct behavior at smaller sizes ($D_p$ < 150 nm) but overestimated the activation at the size range of extrapolated GF-PDFs. Although these uncertainties caused only a small net bias to droplet estimations, they indicate that the estimated GF surfaces may have failed to replicate the real ambient conditions."

Although the minimized residual (Eq. 8) provided some information about the agreement between the two methods, no numerical criteria could be explicitly determined. Instead, the final decisions were made rather relying on personal consideration.

8. **Comment:** *Section 3.5: Specify if the supersaturation is assumed constant or if feedbacks related to the water vapor sink from activated droplets is included.*

**Response:** As we mentioned in our response to comment #1, no feedback effects were included. This piece of information is now provided in the abstract, Sect. 3.5 and summary.

9. **Comment:** *Supplement: Be consistent in the naming of sectors between figure 1 (S1-S5) and table S2 (1-5). Maybe something like WS1-5 could be used, not to confuse with figure labels in the supplement?*

**Response:** Corrected as suggested.

[revised manuscript text omitted]

$$\kappa(D_\mathrm{p}, S_\mathrm{c}) = \frac{4A^3}{27D_\mathrm{p}^3 \ln^2 S_\mathrm{c}}, \tag{5}$$

where $A = 4M_\mathrm{w}\sigma/RT\rho_\mathrm{w}$. Thus, by combining the Eqs. (4) and (5) and by assuming a certain value for $S_\mathrm{c}$, it is possible to estimate the critical growth factor, $GF_\mathrm{c}$, i.e. the required growth factor for particles with dry size $D_\mathrm{p}$ to become activated at the given supersaturation. Thereafter, the size-resolved activation efficiency $f_\mathrm{act,HTDMA}$ can be calculated according to

$$\quad f_\mathrm{act,HTDMA}(D_\mathrm{p}, S_\mathrm{c}) = \int_{GF_\mathrm{c}(D_\mathrm{p},S_\mathrm{c})}^{\infty} \mathrm{GF\text{-}PDF}(GF, D_\mathrm{p})d\mathrm{GF}. \tag{6}$$

Furthermore, the available CCN concentration can  be obtained by weighting the measured particle size distribution with the activation efficiency and by integrating over the whole size range:

$$N_\mathrm{act,HTDMA}(S_\mathrm{c}) = \int_{-\infty}^{\infty} f_\mathrm{act,HTDMA}(D_\mathrm{p}, S_\mathrm{c})\frac{dN_\mathrm{tot}}{d\log D_\mathrm{p}}d\log D_\mathrm{
[revised manuscript text omitted]
}\left(GF, D_p\right) = \frac{dN_{tot}}{d\log D_p}\left(D_p\right) \times \text{GF-PDF}_{tot}\left(GF, D_p\right) - \frac{dN_{int}}{d\log D_p}\left(D_p\right) \times \text{GF-PDF}_{int}\left(GF, D_p\right). \tag{11}$$

Apparently, the relative contribution of less hygroscopic particles is the strongest in the interstitial population and the more hygroscopic mode appears distinctly only in the total and estimated residual aerosol. This is also reflected by the average growth factors. At $D_p$ = 120  and 150 nm, the GFs of cloud droplet residuals are approximately 18% higher than those of interstitial particles. In terms of $\kappa$ values, this discrepancy would correspond to a difference up to 65%–

75% depending on particle size. Again, it should be remarked that similar to activated fractions, the residual GF distributions ($c_{res}$) can appear locally negative if the interstitial concentrations are higher than the respective total concentrations. Before converting these distributions into GF-PDFs shown in Fig. 1, the negative values were set to zeros which eventually strengthened the positive peaks appearing in normalized distributions. As a result, for example the less hygroscopic particle mode appearing in the estimated residual aerosol can be partially attributed to methodological uncertainties.

To our knowledge, this is one of the very few studies characterizing the hygroscopic properties of different in-cloud aerosol populations. Previously, Svenningsson et al. (1994) studied the aerosol hygroscopicity and its relationship to cloud droplet activation at Kleiner Feldberg in Germany. Interstitial aerosol hygroscopicity was measured during cloud events, and by assuming that the air mass was the same, it was compared to the total aerosol sampled during the following clear sky conditions. The less hygroscopic particle fraction was substantially higher in the interstitial population, indicating that the more hygroscopic particles were scavenged into the cloud droplets more efficiently than the less hygroscopic ones. This observation was confirmed in a case study, where a counter flow virtual impactor (CVI) was used to separate the cloud droplets from the total aerosol, so that the hygroscopicity of cloud droplet residuals could be measured independently. Likewise, Rose et al. (2013) observed a decreasing fraction of available CCN during a precipitative cloud event, suggesting that the more hygroscopic particles were mostly activated into cloud droplets and removed from the air through precipitation. Overall, theour results by Svenningsson et al. (1994)from Puijo are in a good agreement with ourthe observations from Puijo.
[revised manuscript text omitted]

---

## Author Comment (AC3) · 13 Jul 2016

**Authors' response to Martin Gysel**

We would like to thank Martin Gysel for his thorough comments and valuable suggestions regarding our manuscript. Based on these comments we have made modifications to our manuscript and slightly changed the analysis method and data presentation. These changes are addressed below with reviewer's comments written in italics and followed by authors' responses (normal font).

*This study investigates the role of aerosol hygroscopicity on its ability to form cloud droplets. Many previous studies have addressed the roles played by particle size and particle hygroscopicity on the ability of such particles to act as cloud condensation nuclei (CCN). However, this was for the most part done based on theoretical considerations, laboratory studies and/or simulated droplet formation on ambient aerosol in CCN counters. This study is one of very few studies that directly investigated the activation of aerosol into the droplets of atmospheric clouds as a function of their size and hygroscopicity. State-of-the-art experimental methods were applied to independently determine the hygroscopicity distributions of the total aerosol and the interstitial aerosol, based on which they could clearly show that the less hygroscopic particles are much less efficient in forming cloud droplets compared to the more hygroscopic particles of equal size. While this essentially confirms expectations, it is still a very valuable result as it directly affects how local/regional emissions contribute to cloud droplet number and to what extent the less hygroscopic particles can be processed by clouds.*

*The manuscript is generally well written, concise and within the scope of ACP. The data analysis approaches seem appropriate for the most part. However, in two cases I am not sure whether the results are internally consistent. Besides, one figure, which is essentially just an internal consistency check of the data analysis approaches, comes as if it was an independent result (see major comments). The "minor comments" are largely just meant to clarify e few things, to improve the notation in equations in order to avoid ambiguities and to provide several ideas for additional analyses. I do not expect that the latter should all be addressed in a comprehensive manner (nor would I expect exhaustive rebuttals for most of them).*

*In conclusion, I recommend this interesting and relevant manuscript for publication in ACP after the most relevant comments have been addressed by the authors.*

**Major comments:**

1. **Comment:** *P.8, l.21-22:"Typically, the activation efficiencies calculated by Eq. (9) ($f_{act,GF>0.80}$) appeared somewhat larger than the DMPS derived values ($f_{act,DMPS}$)." – I claim that this is impossible! It must be caused by errors in the calculations (e.g. from inconsistent choice of GF-PDF normalization approach and integration of the GF-PDFs in Equation 9 or from comparing at different times). Let me explain: "GF>0.80" means that you integrate over the whole GF-range of the GF-PDF. The fact that the GF-PDF, as used in Equation 9, must be normalized to unit area, directly implies that, for "GF>0.80", Equation 9 simplifies to*

   *$f_{act,GF>0.80}(Dp)= ( dN_{tot}(Dp)/dlogDp – dN_{int}(Dp)/dlogDp ) / dN_{tot}(Dp)/dlogDp$*

   *and this is nothing else than how I would define $f_{act,DMPS}(D_p)$. Or in other words: there is no more HTDMA derived information left in $f_{act,GF>0.80}(D_p)$, only DMPS-derived information. Therefore the poor time resolution or other potential issues with the HTDMA measurement cannot influence the*

*result. Please reconsider your calculations and add the results from all other available measurements to Table 1 too, if there was really a mistake in the previous approach.*

**Response:** In the original manuscript, the GF-PDF "scaling factors" were derived from HTDMA measurements, and therefore, all the $f_{act,GF}$ values were independent from DMPS measurements. We agree that this was quite poorly expressed in the text. However, we have re-considered our calculations and decided to change the analysis so that the scaling factors are now derived from DMPS measurements.

As the reviewer pointed out, this should result in equal $f_{act,DMPS}$ and $f_{act,GF>0.80}$ (in the case that $f_{act,GF>0.80}$ is determined by integrating over the whole GF range). However, due to issues with time resolution and relatively slow alteration between the two sampling lines, $f_{act,GF<1.25}$ can sometimes appear negative. Please see our response to comment #2 for further details on this issue.

After revising the calculations, we have decided to include one more cloud event in the manuscript. We also found a small mistake in the analysis as the starting time of event #1 (now event #2) was miscopied as 20:35 instead of 20:25. We have updated Table 1 and revised the related discussion accordingly.

2. **Comment:** *Table 1: Shouldn't the following equality hold for the data shown in Table 1?*

$f_{act,GF>0.80}(D_p) = f_{GF<1.25}(D_p) * f_{act,GF<1.25}(D_p) + (1 - f_{GF<1.25}(D_p)) * f_{act,GF>1.25}(D_p)$

*(simply use Equation 9 to get this). It does seem to be fulfilled for some columns of Table 1 but not for others. Please check your data or clarify the manuscript if I misinterpreted the meaning of Equation 9 or $f_{act}$ or so.*

**Response:** This is a good point and this issue should have been addressed in the manuscript. Ideally, if the activated fraction is zero, the interstitial and total particle concentrations should be equal. However, as the measurements are not done simultaneously, the interstitial concentrations can be occasionally higher than the respective total concentrations. Thus, the activation efficiencies calculated by Eq. (9) (now, Eq. 10) can become negative.

In the original manuscript, these values were reported as zeros but treated as negative when calculating the total activated fractions ($f_{act,GF>0.80}$). By contrast, in the revised version, the negative activation efficiencies are treated as zeros in order to reach consistent values for $f_{act,GF<1.25}$, $f_{act,GF>1.25}$, and $f_{act,GF>0.80}$. On the other hand, this may (again) cause small differences between $f_{act,GF>0.80}$ and $f_{act,DMPS}$.

The following sentences are now added to the manuscript right after the Eq. (10):

"Here it should be noted that the activation efficiencies can appear negative if the averaged interstitial concentrations are higher than the corresponding total values. This can be the case especially within the less hygroscopic regime where the activated fractions are generally low. In such cases, the negative activation efficiencies are reported as zeros and treated as such when calculating the total activated fractions ($f_{act,GF\geq0.80}$). Thus, the resulting $f_{act,GF\geq0.80}$ values can be slightly different from the ones derived solely from DMPS measurements. ".

3. **Comment:** *Figure 3 and first paragraph of Section 3.4:*

*The comparison of cloud droplet predictions made with considering the hygroscopic mixing state or with assuming internal mixture (and using $\kappa_{avg}$) is useful, as it confirms previous results on the sensitivity of CCN predictions to simplified treatment of mixing state for the effect of mixing state of total cloud droplet concentration in real clouds.*

*By contrast, the comparison of $HTDMA+DMPS_{tot}+S_{c,eff}$-predictions of cloud droplet number concentration with those derived from the $DMPS_{tot}$ minus $DMPS_{int}$ measurements is meaningless because it is circular argumentation! Equation 8 was first used to infer the $S_{c,eff}$ from the HTDMA, $DMPS_{tot}$ and $DMPS_{int}$ measurements. The $S_{c,eff}$ obtained in this manner was then used, along with the $HTDMA + DMPS_{tot}$ data, for cloud droplet prediction to be compared with those derived from $DMPS_{tot}$ minus $DMPS_{int}$ measurements as shown in Figure 3. Consequently, this analysis has nothing to do with a closure study, instead it is simply an internal consistency check of the approach to infer $S_{c,eff}$ and to predict cloud droplet number. Therefore, this part of the figure (i.e. the red data points) has to be removed and the discussion needs to be adapted.*

**Response:** Again, this is very good point and we agree that we are not presenting any closure results in Fig. 3. In our opinion, however, this self-consistency check should be included in the paper as it serves a justification/base for the results presented in Sect. 3.5.

In Sect. 3.5, the cloud droplet concentrations are estimated with an assumption that only the more hygroscopic particles were present in the atmosphere. Thereafter, these values are compared to those shown in Fig. 3 (red points). Thus, the main purpose of Fig. 3 is just to assure that the simulated reference values/conditions are comparable to real ambient observations.

Keeping this in mind, we have added the following disclaimer to the manuscript:

> "Nevertheless, since the determination of $N_{act,HTDMA}$ included the estimation of effective peak supersaturation by means of $f_{act,DMPS}$, the comparison shown here can't be considered as a real CCN closure study in an explicit manner. Instead, it rather serves as a base for the analysis presented in the following section (Sect. 3.5)".

**Comment:** *Nevertheless, it is worth to have a closer look into Figure 3. The fact that the grey points have higher correlation and slope closer to unity than the red points is unexpected. If the cloud droplet prediction is so insensitive to assuming internal mixing state, why is then the "internal consistency check" of inferring and re-applying the effective supersaturation from/to the HTDMA measurements so poor (referring to slope and particularly the scatter of the red points)? Is this related to improper treatment of activation plateau values that differ from unity (see separate comment below concerning this potential issue)? Or is the minimization approach applied to infer the effective supersaturation (Equation 8) not suitable? Other reasons?*

**Response:** We apologize for the typo that we had on page 10, line 14. As it was marked in Fig. 3, the line fitted through the red data points had a slope of 1.016 instead on 1.03 (as it was mentioned in the text). It is true, however, that the grey dots had slightly higher $R^2$ than the red points. On the other hand, considering the reasonably low number of data points (n = 26) the difference (0.991 vs. 0.997) is practically negligible and should not be overinterpret.

It is also true that the measured activation curves did not always reach unity at larger sizes, whereas the HTDMA derived curves typically did. Therefore, it is definitely one possible explanation for the positive biases between $N_{act,HTDMA}$ and $N_{act,DMPS}$ ($N_{act,HTDMA} > N_{act,DMPS}$). We also tried an alternative minimization approach by simply minimizing the difference between the HTDMA and DMPS derived D50s, but the effect was negligible.

**Minor comments:**

4. **Comment:** *Abstract: Well written but for one missing element: The range of supersaturations occurring in the is not mentioned. This would be worthwhile as they appear to be rather low, which increases the sensitivity of CCN number concentration to hygroscopicity/composition compared to high supersaturation.*

   **Response:** The range of effective peak supersaturation is now included in the abstract.

5. **Comment:** *P.3, l.26: It might be worthwhile to mention how the visibility was measured.*

   **Response:** The weather sensors are now described in the manuscript.

6. **Comment:** *Sect. 2.3: A $PM_{1.0}$ impactor was used in the interstitial inlet to remove the activated cloud droplets just leaving the interstitial aerosol behind. However, the mode of none-activated droplets in stable equilibrium within the cloud may potentially extend to diameters larger than 1 μm, if the peak supersaturation at cloud formation was very low (the lower the peak supersaturation, the larger the activation cut-off diameter, the larger the maximal diameter of non-activated interstitial particles). In such a case, the largest and most hygroscopic particles of the true interstitial aerosol in the cloud would be missing in the aerosol sample measured behind the interstitial inlet. Hammer et al. (2014) address this issue in more detail. Can you exclude this potential artefact based on supporting data such as droplet size distributions or estimates of the maximal possible equilibrium droplet size for the clouds formed with the lowest peak supersaturation?*

   *Hammer, E., Gysel, M., Roberts, G. C., Elias, T., Hofer, J., Hoyle, C. R., Bukowiecki, N., Dupont, J.C., Burnet, F., Baltensperger, U., and Weingartner, E.: Size-dependent particle activation properties in fog during the ParisFog 2012/13 field campaign. Atmos. Chem. Phys., **14**, 1051710533, doi:10.5194/acp-14-10517-2014, 2014.*

   **Response:** This is a very good point and definitely something that we need to take into account in the future measurements. At this stage, however, we can neither exclude, nor quantify, the possible uncertainties inflicted to our observations.

7. **Comment:** *Sect. 2.4: The residence time between humidifier and DMA is provided (2s). The residence time within the humidifier might also be of interest, possibly even including a brief remark on the reasons for choosing a rather short or rather long residence time.*

**Response:** The residence time inside the humidifier was approximately 0.2 s. This piece of information is now added to the manuscript.

8. **Comment:** *Equation 2 and other instances in Sect. 2.5 and possibly the rest of the manuscript: This equation (Köhler-equation) describes the equilibrium water vapour saturation ratio ($S_{eq}$) as a function of particle hygroscopicity, diameter, etc. By contrast, the critical saturation ratio ($S_c$) is the maximum of the Köhler curve described with Equation 2. You are using the same symbol for these two different quantities. This needs to be fixed.*

   **Response:** Thank you for pointing out this mistake. $S_c$ in Eq. (2) is now changed to $S_{eq}$ and determined as an equilibrium saturation ratio. Later, $S_c$ is defined as a critical saturation ratio and explained accordingly.

9. **Comment:** *Line just below Equation 3, "...where $V_s$ and $V_w$ are the soluble and water volumes...": In general, $V_s$ is the volume of the whole dry particle, not just the soluble components. Insoluble fractions, if present, are accounted for with the κ-value.*

   **Response:** Good point. We have replaced $V_s$ with $V_p$ and rephrased the following definition as "…where $V_p$ *and* $V_w$ are the dry particle and water volumes…"

10. **Comment:** *Equation 5: This analytical solution for the critical saturation ratio contains mathematical approximations which become increasingly inaccurate with decreasing critical saturation ratio. Therefore, this equation should only be used for qualitative purposes, whereas a numerical solution of the Köhler equation must be implemented for quantitative purposes.*

   **Response:** This is also a good point, and therefore, Eq. (5) is now referred as an approximation. In addition, we repeated the simulations by solving the Köhler equation numerically. However, the effect was negligible and most likely compensated by very minor changes in the estimated $S_{c,eff}$ values.

11. **Comment:** *Equation 5 and line just below: This is now the critical saturation ratio. It should be explained what it is – in contrast to the equilibrium saturation ratio appearing in Equation 2 – and the symbol should be defined.*

   **Response:** Please see our response to comment 8.

12. **Comment**: *Equations 6 and 7: It could be mentioned that the approach and equations used to predict the CCN number concentration from particle number size distribution (from SMPS) and GF-PDFs (from HTDMA) is identical to the approach introduced by Kammermann et al., 2010b (cf. their Equations 2&3).*

   **Response:** Reference to Kammermann et al. (2010b) is included.

13. **Comment:** *P.6, l. 5‑8: This step also involves interpolation in time (besides inter‑/extrapolation in size).*

    **Response:** Yes. Or alternatively, temporal averaging.

14. **Comment:** *P.6, l. 11ff: "Nonetheless, it has to be noted that above 200 nm, hygroscopicity is quite rarely a limiting factor and the most crucial activation characteristics are dependent on particle properties between the ~80 nm and ~200 nm sizes." – Isn't this statement somewhat in conflict with your result that a substantial portion of the non‑hygroscopic particles remains interstitial (at the largest diameter covered by your measurements)? Therefore you need the additional argument that the number fraction of non‑hygroscopic particles depends only weakly on size across the size range relevant in this context.*

    **Response:** This part of the text was indeed unclearly written. What we really wanted to state here was that according to the observed size dependence of $f_{\text{GF}<1.25}$, the particles at $D_\text{p} > 200$ nm were supposedly characterized by reasonably low number fractions of less hygroscopic particles. As a result, small uncertainties in the estimated GF-PDFs would only cause a small net bias to droplet predictions.

    Nevertheless, as we don't have actual measurement data for this size range (and this argument would be limited to average conditions at Puijo), we have decided to omit this statement from the manuscript.

15. **Comment:** *P.6, l.13ff: "Secondly, the method assumes that the subsaturated hygroscopicities are representative for supersaturated conditions. Such an assumption is not always totally valid and discrepancies between the two saturation regimes have been reported based on laboratory and field experiments…" – The study by Jurányi et al. (2013) could also be referenced here, as one of the examples that found very good closure between sub‑ and supersaturated regimes for externally mixed urban aerosol.*

    **Response:** We have added a reference to Jurányi et al. (2013).

16. **Comment:** *P6., l.19ff: Comments on the approach to estimate the effective peak supersaturation (PS: I'd suggest $S_{c,eff}$ rather than $S_{eff}$ as symbol):*

    **Response:** $S_\text{eff}$ and $s_\text{eff}$ are replaced with $S_{\text{c,eff}}$ and $s_{\text{c,eff}}$ as suggested.

    a. *I suggest to start with a brief explanation of the concept behind estimating $S_{c,eff}$, possibly also referring to Hammer et al. (2014). This would be likely be helpful for the "average" reader of this manuscript.*

       **Response:** We have included a brief definition of $S_{\text{c,eff}}$ as well as a reference to Hammer et al. (2014).

**b.** *If entrainment occurs or in the case of partially/fully glaciated clouds the plateau value of $f_{act,DMPS}$, i.e. the value $f_{act,DMPS}$ takes at large diameters at which also the nonhygroscopic particle activate, may be substantially smaller than unity (e.g. Fig. 3 in Verheggen et al., 2007). The minimization approach given in Equation 8 would cause a bias for such a scenario. – Did you observe evidence for entrainment and/or glaciation or did the plateau value of $f_{act,DMPS}$ always reach unity?*

*Verheggen, B., Cozic, J., Weingartner, E., Bower, K., Mertes, S., Connolly, P., Gallagher, M., Flynn, M., Choularton, T., and Baltensperger, U.: Aerosol partitioning between the interstitial and the condensed phase in mixed-phase clouds. J. Geophys. Res., **112**, D23202, doi:10.1029/2007JD008714, 2007.*

**Response:** As we mentioned in our response to comment #3, the DMPS derived activation curves did not always reach unity. Thus, we can't fully exclude the possibility of entrainment. On the other hand, the reduced activation efficiency of larger particles was also associated with reasonably low particle concentrations ($\rightarrow$ small deviations in particle concentrations may cause relatively large fluctuations in number fractions). Thus, distinguishing the effect of entrainment from measurement uncertainties/biases can be tricky.

In order to minimize the possible biases arising from minimization, the residual (Eq. 8) was calculated over the 80–200 nm size range.

**c.** *Our experience from similar measurements at the Jungfraujoch research station is that the diameter range across which $f_{act,DMPS}$ increases from 0 to 1 is much broader than can be explained with the heterogeneity of the aerosol in terms of mixing state/GF-PDF (the JFJ-aerosol is rather internally mixed). This indicates that the width of $f_{act,DMPS}$ is mainly driven by heterogeneity of $S_{c,eff}$ on small spatial scales due to e.g. turbulence. What does it look like in your case (the aerosol at Puijo is obviously much more externally mixed than that observed at the Jungfraujoch)? Can the shape of $f_{act,DMPS}(D_p)$ be explained with the external mixing alone? This question possibly goes beyond the main focus of this paper, but it might still be worth looking at it. You have the data at hand and could possibly produce a supplementary figure using representative examples).*

**Response:** One criteria for case selection in Sect. 3.4 (and Sect. 3.5) was that the $f_{act,HTDMA}$ managed to produce the size-dependence/slope similar to $f_{act,DMPS}$. Ideally (assuming that the GF-surfaces were well-predicted), this would indicate that the changes in hygroscopicity/mixing state were enough to explain the width of $f_{act,DMPS}$.

However, as pointed out in the revised manuscript, this wasn't always the case and thus, we can't exclude the possible influence of turbulent/heterogeneous $S_{c,eff}$. This is definitely an interesting topic but would require a much more comprehensive set of analysis to be addressed adequately.

**d.** *An alternative approach would be to fit something like a sigmoid curve into $f_{act,DMPS}(D_p)$ to obtain an effective activation cut-off diameter (half rise). Inserting this cut-off diameter and $\kappa_{avg}(D_p)$ into Köhler theory then provides $S_{c,eff}$ under the assumption of internal mixing. I would expect that these values are very similar to those obtained with your approach accounting for external mixture. If not, you should comment on the fact that proper treatment of mixing state is crucial for inferring $S_{c,eff}$ when dealing with clouds formed on externally mixed aerosol.*

**Response:** This is also a good idea. However, determination of cloud supersaturation is likely beyond the main scope of the current manuscript and will be examined more comprehensively in future studies.

e. *Did you define $f_{act,DMPS}(D_p)$?*

**Response:** The definition of $f_{act,DMPS}$ is added to the manuscript.

17. **Comment:** *P.6, l.26ff: First, this paragraph belongs above the paragraph describing how you infer $S_{c,eff}$, as this is still about predicting CCN number concentration from HTDMA data, if I got that right. Second, you may have to explain how you obtain $\kappa_{avg}(D_p)$.*

**Response:** We have decided to keep the order of these paragraphs as they are. The main idea here is that adjusting the $S_c$ is an integral part of determining $N_{act,HTDMA}$, rather than an individual analysis. Another reason is that the obtained $S_{c,eff}$ values are later used in the internally mixed approach.

We have rephrased the paragraph describing the internal mixing approach to improve the language and clarity.

18. **Comment:** *Sect. 3.1: Some at least partially glaciated clouds wouldn't be surprising if ambient temperature was sometimes below zero (minimum was -9.7 °C).*

**Response:** The minimum temperature during cloud events was -5.8 °C. Although we do not have direct measurements on ice nuclei activity, our current perception is that glaciation plays a very minor role at these temperatures (at Puijo).

19. **Comment:** *Figure 1, bottom row: GF-PDFs shown here are normalized to unit area. I wonder whether it would be instructive to add an extra row (or replace the current bottom row) with a version in which you re-normalize the GF-PDFs as follows. For the total aerosol, multiply the GF-PDF that is already normalized to unit area with $dN_{tot}/dlogD_p(D_p)$. And equivalently for the interstitial aerosol. The area between the curves representing the total and interstitial GF-PDFs for equal size would then directly correspond to the activated particles. Furthermore, normalized in this manner, the bottom row of Figure 1 would then be are more close graphical representation of what you calculate with Equation 9. You could even add an extra row of panels that shows $f_{act,GF}$ for each diameter and GF-resolved, i.e. as a function of GF rather than integrated over a GF range.*

**Response:** Due to highly variable total/interstitial concentrations, averaging these kind of "non-normalized" GF distributions would result in very different looking distributions compared to what is presented in the current version of Fig. 1 (i.e., the averaged distributions would be largely biased towards the observations with high concentrations). To avoid any confusion, we've decided to keep the Fig. 1 as it is and include the suggested figure in the supplementary material. In this figure, shown are the median GF distributions and their 25[th] and 75[th] percentiles determined from hourly averaged HTDMA+DMPS observations.

20. **Comment:** *P.7, l.20: Was the shift of the more hygroscopic mode towards larger GF with increasing particles size less or more than what can be explained by the size dependence of the GF imposed by the Kelvin effect?*

**Response:** Typically, the $\kappa$ values of more hygroscopic particles increased with particle size indicating that the Kelvin effect alone was not enough to explain the observed shift towards larger growth factors. This can be seen from Fig. 1 (see the attached figure below) showing the average $\kappa$-PDFs calculated over the whole measurement campaign. This figure is also included in the manuscript together with a brief explanation.

[Figure]

**Fig. 1:** Mean $\kappa$-PDFs of 80, 120 and 150 nm particles averaged over the whole campaign. The shaded areas represent the ranges between the 25[th] and 75[th] percentiles.

21. **Comment:** *P.7, l21ff: The observations by Laborde et al. (2013) in Paris revealed even a little more detail. There was very clear evidence that particles from fresh traffic emissions appeared mainly at GF≈1.0, whereas particles from wood burning appeared mainly at GF≈1.1, together forming the "nonhygroscopic" mode in the HTDMA. I have added this detailed remark because the GF-PDFs shown in the bottom row of your Figure 1 for particles with diameters of 120 nm and 150 nm seem to provide evidence that the cloud droplet active fraction differs slightly between GF≈1.0 and GF≈1.1. Could you confirm this or is this difference within uncertainty?*

**Response:** This is a good point. As we mentioned in our response to comment #2, the difference between the interstitial and total concentrations ("TOT-INT") could become negative if the activated fraction was low. When estimating the residual GF-PDFs, these values are first set to zeros, which leads to a strengthening of positive peaks when the distributions are normalized. Therefore, the difference between GF = 1 and GF = 1.1 is at least partially an artefact, rather than a real difference in activation efficiency. This is now explained in the manuscript.

**22. Comment:** *Equation 9: $N_{tot}(D_p)$ should be replaced by $dN_{tot}(D_p)/dlogD_p$, shouldn't it? And so for $N_{int}(D_p)$? – You define $N_{tot}$ as: "…where $N_{tot}$ and $N_{int}$ are the total and interstitial number concentrations…". This rather rather sounds as if $N_{tot}(D_p)$ was representing a cumulative number concentration, which would be wrong in Equation 9 (as I understand the purpose of Equation 9).*
*PS: you could of course also choose $dN_{tot}(D_p)/dD_p$ instead of $dN_{tot}(D_p)/dlogD_p$ as the factor in between those two eventually cancels out.*
*Besides: I recommend adding another line to Equation 9, in which you rearrange it as follows: fact,GF1<GF<GF2(Dp, GF1, GF2)= ( dNtot(Dp)/dlogDp * f tot,GF1<GF<GF2(Dp) – dNint(Dp)/dlogDp *f int,GF1<GF<GF2(Dp) ) / dNtot(Dp)/dlogDp *f tot,GF1<GF<GF2(Dp) where $f_{tot,GF1<GF<GF2}(D_p)$ is the number fraction of particles (total aerosol) with dry diameter $D_p$ and GF between GF1 and GF2 (and equivalent for the interstitial particles). This addition should help in understanding the meaning of Equation 9.*

**Response:** We have modified the equation as suggested.

**23. Comment:** *P.9, l.3ff: "The most interesting remark concerns the difference between the low and high hygroscopicity particles at 120 and 150 nm. While the activation efficiency of total aerosol and more hygroscopic particles increases with size, the less hygroscopic particle mode remains almost non-activated." – The "size dependence" mentioned in this statement is distracting from the main message. In my view Table 1 already captures the central and very nice results of your study, which is: "…., the cloud droplet activated fraction of the less hygroscopic particles is much smaller than that of the more hygroscopic particles of equal size…which confirms that cloud droplet activation critically depends on particle hygroscopicity for particle sizes for all sizes in the range of the droplet activation cut-off…." The very nice thing is that you showed this, which is expected based theory and hygroscopicity-resolved HTDMA-CCN closure studies, for the activation of atmospheric aerosols in atmospheric clouds. Personally I would focus on this, i.e. comparing less versus more hygroscopic at equal size, and address size dependence in the next paragraph.*

**Response:** This part of the manuscript has been re-written according to the updated results.

**24. Comment:** *P.9, l.11ff: "Here, the residual aerosol-properties were estimated indirectly by using the hourly averaged total and interstitial GF-PDFs and their actual number concentrations." – Which factors did you apply to the normalized GF-PDFs, $dN_{tot}(D_p)/dlogD_p$ and $dN_{int}(D_p)/dlogD_p$ for the total and interstitial inlets or did you revert the normalization factor of the GF-PDFs with the normalization factor that had been applied? I would believe that the former is the better choice, if the total and interstitial DMPS measurements are corrected such that they are identical for out of cloud measurements. However, this is just a subtlety.*

*PS: Applying the number of counts of the HTDMA raw measurements would be "wrong" because the detection probability in the HTDMA is GF-dependent for a fixed dry size. However, again just a small but still systematic bias.*

**Response:** In the original manuscript, the GF-PDFs were scaled by using the concentrations derived from HTDMA measurements. In the current version, we are instead using the DMPS derived concentrations. This now clarified in the manuscript.

25. **Comment:** *P.9, l.10-17: why do you not discuss the size dependence of the activated fractions of the more and of the less hygroscopic particles in this paragraph? There are good reasons for how they depend on size and why there is hardly any difference between total and interstitial inlet at the smallest covered size.*

    **Response:** We have added a couple of sentences regarding the observed size dependence.

26. **Comment:** *Figure 1: According the legend in the bottom row of Figure 1 the difference of the average GF between total and interstitial inlet is 0.04 for the two dry diameters 80 nm and 120 nm. However, the difference between total and interstitial seems to be much larger for 120 nm compared to 80 nm when looking at the bottom row of Figure 1. Please check carefully and adapt the figure and discussion on P.9 l.10-17 if needed*

    **Response:** Thank you for pointing this out. There was indeed a mistake in the legend values in the case of interstitial aerosol. The figure is updated with revised values and the discussion is adapted.

27. **Comment:** *P.9, l.18 ff: "To our knowledge, this is one of the very few studies characterizing the hygroscopic properties of different in-cloud aerosol populations." – there might exist some CCN based literature on this topic; you could check for authors like U. Pöschl and D. Rose. Concerning chemical composition: you could check for SP2-based studies by J. Schroder et al. This might potentially link in to the behaviour of the less-hygroscopic particles.*

    **Response:** We have added a reference to discussion paper by Rose et al. (2013).

28. **Comment:** *Figure 2 and associated discussion: Relevant analysis, however, somewhat incomplete. Equation 5 tells us that the three parameters $S_c$, $\kappa$ and $D_{50}$ are related to each other. Therefore, the relation between the three of them should be reflected in the analysis, figures and discussion. Some thoughts on this:*

    a. *The dependence of $D_{50}$ on $\kappa$ could be the result of cross-correlation rather than causality. You should confirm that $\kappa$ and $S_c$ are not correlated to make your result stronger. This is definitely required before you make the statement at the end of Section 3.3.*

       **Response:** No correlation was found between $\kappa$ and $S_{c,eff}$ ($R^2 \sim 0.02$). This is now mentioned in the manuscript.

    b. *Color code: the most relevant information I seem to learn from the colour code is that the variability of $\kappa$ is for the most part driven by the variability of the number fraction of less and more hygroscopic particles rather than the variability of the respective mean GFs of these two modes. Correct? This would be better seen from a scatter plot of $\kappa$ versus $f_{GF<1.25}$.*

**Response:** This essentially true. According to simple linear correlation, the less hygroscopic fraction explained approximately 80% of the total variation in $\kappa$ regardless of particle size. On the other hand, the less hygroscopic fraction is also linked to the position of the more hygroscopic mode ($R^2 \sim 0.3$–$0.4$), which makes the comparison a bit more complicated.

c. *Fit curve: the fit curve can be quantitatively interpreted, i.e. it provides you an estimate about the $S_{c,eff}$ "averaged" over the whole data set. Is this value consistent with your other analyses of $S_{c,eff}$? Caveat: Equation 5 is an approximation, which is likely not accurate for the rather low critical supersaturations you are dealing with.*

**Response:** This estimation is given in Sect. 3.4.

d. *I suggest you include multiple theoretical lines in Figure 2 that show $D_{50}$ versus $\kappa$ for different $S_c$ (based on unbiased numerical solutions rather than the approximate Equation 5). These theoretical lines might possibly save you the trouble of including a fit curve. Additionally you should choose the $S_{c,eff}$ as colour scale for the data points (you can have multiple versions with different colour scales if you like to keep your old colour scale too). This will give you a more complete picture on the influence of $\kappa$, $S_{c,eff}$ and also "measurement noise" on the variability of $D_{50}$.*

**Response:** We have added theoretical lines for critical supersaturations of 0.10, 0.15, 0.20, 0.30, 0.40 and 0.50%. However, the suggested figure (attached below) is included in the supplementary material. The reason is that due to the discrepancies between the HTDMA and DMPS derived activation properties (highlighted in Sect. 3.4), some of the determined $s_{c,eff}$ values were most likely characterized by increased uncertainties.

[Figure]

e.  *You should also create a figure in which you swap the roles of κ and $S_{c,eff}$, i.e. you plot $D_{50}$ vs $S_{c,eff}$ and choose $κ_{avg}$ (I'd say for 120 nm or 150 nm or a value interpolated to the mean D50) as colour scale (theoretical lines should also be added). How does it compare with the figure suggested above?*

    **Response:** This figure is also included in the supplementary material.

[Figure]

f.  *The outlier in Figure 2: is it an outlier in the sense of "cannot be explained" or do you have independent evidence that the very high $D_{50}$ could possibly be caused by exceptionally low supersaturation (you cannot use $S_{c,eff}$ to argue as $S_{c,eff}$ is inferred from D50)?*

    **Response:** In this case, the D50 was most likely biased towards larger sizes due to the uncertainties arising from extremely low number concentrations (i.e., the activation curve wasn't perfectly smooth around ~170–240 nm).

29. **Comment:** *P.10, l.28-29: "…the estimated peak supersaturations… …they provide some valuable information about the in‑cloud conditions…". – The droplet activation happens at the initial stages of cloud formation.*

    **Response:** This paragraph is rephrased as "…about the conditions relevant to cloud droplet formation".

30. **Comment:** *Last paragraph of Section 3.4 (comparison of supersaturations with literature): Hammer et al. (2014) reported a systematic difference in observed peak supersaturations for the two prevalent wind directions, which could be explained by differences in the orographic forcing (steep vs gentle mountain slopes). What are the cloud formation mechanisms for the clouds probed at Puijo (and Pallas)? Are the lower peak supersaturations at those two sites possibly related to weaker orographic forcing compared to the Puy de Dôme and Jungfraujoch sites?*

    **Response:** We are currently preparing a manuscript concentrating on the effect of updrafts on cloud properties at Puijo. The preliminary model results suggest that due to the reasonably low height of the Puijo hill and the fact that the measurement station is located on the top of a 75-metre tower, the terrain topography has a minor effect on the observed cloud properties. We have slightly widened the discussion regarding the results by Hammer et al. (2014).

**31. Comment:** *Figure 4 and associated discussion: The susceptibility of cloud droplet concentration to hygroscopicity can be quite asymmetric with respect to increase vs decrease of κ (see e.g. Figure 8 in Juranyi et al., 2010, or other studies that did similar sensitivity analyses for CCN number concentrations). Instead of just considering the case "no less hygroscopic particles at all" (higher κ), you could additionally consider the case "no more hygroscopic particles at all" (lower κ) for the sensitivity analysis presented in your Figure 4 and Table 2.*

*Jurányi, Z., Gysel, M., Weingartner, E., DeCarlo, P. F., Kammermann, L., and Baltensperger, U.: Measured and modelled cloud condensation nuclei number concentration at the high alpine site Jungfraujoch. Atmos. Chem. Phys., **10**, 7891-7906, doi:10.5194/acp-10-7891-2010, 2010. PS: further down in the manuscript it became clear why you specifically look at positive deviations. You could try to clarify this earlier.*

**Response:** Actually, we already performed this kind of an analysis when preparing the manuscript. In our opinion, however, including several scenarios provided only little added value considering the main message of our manuscript.

Anyhow, in case you're interested, please see the attached figure showing the results from simulations where the less hygroscopic GF modes were rescaled by a positive factor of 2.5 (i.e. the less hygroscopic fractions were increased by 150% → hygroscopicities decreased).

[Figure]

**32. Comment:** *Concerning difference of the activation behaviour of the two modes: As the aerosol at Puijo appears to have two rather well separated hygroscopicity modes, and since you prove that this directly affects the cloud droplet formation ability, you could quantify the expected difference of activation cut-off diameter for these two modes, if you like. One option would be the following: from "every" HTDMA measurement you can infer $S_{c,eff}$, $\kappa_{avg,GF<1.25}$ and $\kappa_{avg,GF>1.25}$. This allows to infer $D50,GF<1.25$ and $D50,GF>1.25$. Plotting $D50,GF<1.25$ and $D50,GF>1.25$ versus $S_{c,eff}$ then gives a fair idea of the activation cut-off diameter of the two modes, which is for example relevant for the threshold size down to which the particles in either mode can undergo cloud processing under the conditions in clouds at Puijo. (The only thing you would have to think about is how to deal with the diameter depends of hygroscopicity.)*

**Response:** A relevant analysis but most preferably to be included in future studies.

**33. Comment:** *There is another potentially interesting question you could look at if you like: while assuming an internally mixed aerosol can provide very good estimates of the total CCN number concentration, if properly done, it will not give an accurate answer concerning the respective contributions of the background aerosol and local/regional emissions to CCN number (with the picture in mind that the less hygroscopic mode is of local/regional origin). Based on your data set you could make at least a crude estimate of how the number fraction of local/regional particles compares between total aerosol and those particles that formed cloud droplets (pulling the idea of the previous comment even a little further). Or in other words: your data set seems to imply that most particles of local/regional origin have to undergo quite some atmospheric aging processes until they start participating in cloud droplet formation, doesn't it?*

**Response:** This is also a very good idea and well-linked to our ongoing work concentrating on the influence of nearby pollution sources.

**34. Comment:** *l.26-27: "Understandably, by suppressing the size-dependent variations in chemical composition, the activation curves become steeper and the D50s decrease." – This is unclear. You only present data from a single size, so how can size dependence be suppressed? To my understanding the D50s decrease because you make the particles more hygroscopic! Please clarify how you mean this.*

**Response:** The idea here is that neglecting the effect of less hygroscopic particles suppresses the size dependence of hygroscopic growth factors (that is, the average GFs increase much more clearly with particle size when the less hygroscopic mode in included – please see Table 2).

As we are not changing the effective peak supersaturation, the activation starts to occur approximately at same sizes in both scenarios (original mixing state vs. high hygroscopicity assumption). However, as the particles become more hygroscopic the shape of the activation curves changes, i.e. the curves become steeper, which decreases the D50s. We have tried to use more precise wording here.

**Technical corrections:**

**35. Comment:** *P.7, l24: In the context of HTDMA measurements I would speak of "non-hygroscopic" particles for GF=1.0 rather than hydrophobic. I'd rather use the latter term in the context of measurements that are sensitive to "wettability", i.e. adsorption or contact angle or similar.*

**Response:** Corrected as suggested

**In-cloud measurements highlight the role of aerosol hygroscopicity in cloud droplet formation**

Olli Väisänen[1], Antti Ruuskanen[2], Arttu Ylisirniö[1], Pasi Miettinen[1], Harri Portin[3], Liqing Hao[1], Ari Leskinen[1,2], Mika Komppula[2], Sami Romakkaniemi[2], Kari E. J. Lehtinen[1,2], Annele Virtanen[1]

[1]University of Eastern Finland, Department of Applied Physics, P.O. Box 1627, 70211 Kuopio, Finland
[2]Finnish Meteorological Institute, P.O. Box 1627, 70211 Kuopio, Finland
[3]Helsinki Region Environmental Services Authority, P.O. Box 100, 00066 HSY, Finland

*Correspondence to*: A̶.̶Annele Virtanen (annele.virtanen@uef.fi)

**Abstract.** The relationship between aerosol hygroscopicity and cloud droplet activation was studied at the Puijo measurement station in Kuopio, Finland, during the autumn 2014. The hygroscopic growth of 80, 120 and 150 nm particles was measured at 90 % relative humidity with a hygroscopic tandem differential mobility analyzer. Typically, the growth factor (GF) distributions appeared bimodal with clearly distinguishable peaks around 1.0–1.1 and 1.4–1.6. However, the relative contribution of the two modes appeared highly variable reflecting the v̶a̶r̶y̶i̶n̶g̶probable presence of fresh anthropogenic particle emissions. The hygroscopicity-dependent activation properties were estimated in a case study comprising t̶h̶r̶e̶e̶four separate cloud events with varying characteristics. At 120 and 150 nm, the activation efficiencies within the low- and high-GF modes varied between 0 0̶.̶3̶3̶%–34% and 0̶.̶6̶6̶–̶0̶.̶8̶6̶,̶57%–83%, respectively, indicating that the less hygroscopic particles remained a̶l̶m̶o̶s̶t̶mostly non-activated, whereas the more hygroscopic mode was predominantly scavenged into cloud droplets. By modifying the measured GF distributions, it was estimated how the cloud droplet concentrations would change if all the particles belonged to the more hygroscopic group. According to t̶h̶e̶ $\kappa$-Köhler simulations, the cloud droplet concentrations increased up to 70 %̶ ̶w̶i̶t̶h̶ ̶i̶n̶c̶r̶e̶a̶s̶i̶n̶g̶ ̶h̶y̶g̶r̶o̶s̶c̶o̶p̶i̶c̶i̶t̶y̶% 
[revised manuscript text omitted]
(D_{\mathrm{p}}, S_{\mathrm{c}}) = \frac{4A^3}{27D_{\mathrm{p}}^3 \ln^2 S_{\mathrm{c}}}, \tag{5}$$

where $A = 4M_{\mathrm{w}}\sigma/RT\rho_{\mathrm{w}}$. Thus, by combining the Eqs. (4) and (5) and by assuming a certain value for $S_{\mathrm{c}}$, it is possible to estimate the critical growth factor, $\mathrm{GF_c}$, i.e. the required growth factor for particles with dry size $D_{\mathrm{p}}$ to become activated at the given supersaturation. Thereafter, the size-resolved activation efficiency $f_{\mathrm{act,HTDMA}}$ can be calculated according to

5 $$f_{\mathrm{act,HTDMA}}(D_{\mathrm{p}}, S_{\mathrm{c}}) = \int_{\mathrm{GF_c}(D_{\mathrm{p}}, S_{\mathrm{c}})}^{\infty} \mathrm{GF\text{-}PDF}(\mathrm{GF}, D_{\mathrm{p}}) d\mathrm{GF}. \tag{6}$$

Furthermore, the available CCN concentration can  be obtained by weighting the measured particle size distribution with the activation efficiency and by integrating over the whole size range:

$$N_{\mathrm{act,HTDMA}}(S_{\mathrm{c}}) = \int_{-\infty}^{\infty} f_{\mathrm{act,HTDMA}}(D_{\mathrm{p}}, S_{\mathrm{c}}) \frac{dN_{\mathrm{tot}}}{d\log D_{\mathrm{p}}} d\log D_{\mathrm{
[revised manuscript text omitted]
_{\text{act}}(D_p, \text{GF}_1, \text{GF}_2) = \frac{\int_{\text{GF}_1}^{\text{GF}_2}[N_{\text{tot}}(D_p) \times \text{GF-PDF}_{\text{tot}}(\text{GF},D_p) - N_{\text{int}}(D_p) \times \text{GF-PDF}_{\text{int}}(\text{GF},D_p)]\,d\text{GF}}{\int_{\text{GF}_1}^{\text{GF}_2} N_{\text{tot}}(D_p) \times \text{GF-PDF}_{\text{tot}}(\text{GF},D_p)\,d\text{GF}},\qquad(9)$$

$$f_{\text{act},\text{GF}_1<\text{GF}<\text{GF}_2}(D_p) = \frac{\frac{dN_{\text{tot}}}{d\log D_p}(D_p) \times f_{\text{tot},\text{GF}_1<\text{GF}<\text{GF}_2}(D_p) - \frac{dN_{\text{int}}}{d\log D_p}(D_p) \times f_{\text{int},\text{GF}_1<\text{GF}<\text{GF}_2}(D_p)}{\frac{dN_{\text{tot}}}{d\log D_p}(D_p) \times f_{\text{tot},\text{GF}_1<\text{GF}<\text{GF}_2}(D_p)},\qquad(10)$$

where $f_{\text{tot,GF1<GF<GF2}}$ and $f_{\text{int,GF1<GF<GF2}}$ were the total and interstitial number fractions of particles with dry size $D_p$ and GF between GF$_1$ and GF$_2$ . The average activation efficiencies were calculated

10  separately for each cloud event and for three different GF regimes (GF $\geq$ 0.80, 0.80 $\leq$ GF < 1.25 and GF $\geq$ 1.25). Here it should be noted that the activation efficiencies can appear negative if the averaged interstitial concentrations are higher than the corresponding total values. This can be the case especially within the less hygroscopic regime where the activated fractions are generally low. In such cases, the negative activation efficiencies are reported as zeros and treated as such when calculating the total activated fractions ($f_{\text{act,GF}\geq0.80}$). Thus, the resulting $f_{\text{act,GF}\geq0.80}$ values can be slightly different from those derived solely

15  from DMPS measurements.
A total of nine cloud events were observed during the twin inlet period. Due to the reasonably low time resolution of HTDMA size scans~~, which can lead to biased results if the changes in the air masses are not captured

20  adequately by both of the sample lines. Therefore, we only present the results from threewith reasonably good agreement between the two methods.three cloud eventdata isnm~~and 150 nm sizes.

25  The duration of the selected cloud events varied from 1 h 31 min (event #2) to 4 h 25 min (event #3 the longest.

4). Cloud event #1 had

 the
highest particle and
cloud droplet number concentrations, up to 2935 cm$^{-3}$ and 781 cm$^{-3}$, respectively, whereas
much lower values were observed during the latter events. For example during the cloud event #4, the particle and cloud
droplet concentrations were down to 792 cm$^{-3}$ and 69 cm$^{-3}$, respectively. These four cloud events were also characterized by
very different wind patterns. Event #2 was influenced by westerly winds blowing across the clean sector. By contrast, the wind
was from the northeast during cloud event #
3 and from the southeast during the cloud
 event #4. Furthermore, event #1 was dominated by southwesterly winds blowing
across the transition region between the clean and polluted sectors.
The most interesting remark, however, concerns the different activated fractions within the two GF modes. At $D_p$ = 120 nm,
the activation efficiencies of less hygroscopic particles varied from 0% to 4%, whereas the values for more hygroscopic
particles were much higher (between 57% and 70%). A similar trend was observed also
at $D_p$ = 150 nm, with corresponding intervals of 0%–34% and 78%–83%, respectively.
One may also note that the hygroscopicity-dependent activation efficiencies increased with particle size. In the case
 of more hygroscopic particles, this was most likely attributed to Kelvin effect and small increments in respective
hygroscopicities. Besides, the cloud events #2 and #4 were characterized by somewhat increased $f_{act,GF<1.25}$
values at $D_p$ = 150 nm. Although it is possible that part of these less hygroscopic particles were scavenged into
cloud droplets, it's good to note that these cases were also characterized by notably low particle concentrations which may
have led to increased uncertainties in activated fractions.
The difference between the activated and non-activated particles is also illustrated in Fig. 1 (lower panel) where the average
GFs and GF-PDFs are presented separately for total, interstitial and residual aerosol populations. Here, the residual aerosol
properties were estimated indirectly by using the hourly averaged total and interstitial GF-PDFs and the
respective ambient particle concentrations as follows:

$$c_{res}\left(GF, D_p\right) = \frac{dN_{tot}}{d \log D_p}\left(D_p\right) \times \text{GF-PDF}_{tot}\left(GF, D_p\right) - \frac{dN_{int}}{d \log D_p}\left(D_p\right) \times \text{GF-PDF}_{int}\left(GF, D_p\right). \quad\quad (11)$$

Apparently, the relative contribution of less hygroscopic particles is the strongest in the interstitial population and the more
hygroscopic mode appears distinctly only in the total and estimated residual aerosol. This is also reflected by the average
growth factors. At $D_p$ = 120  and 150 nm, the GFs of cloud droplet residuals are approximately 18% higher than those
of interstitial particles. In terms of $\kappa$ values, this discrepancy would correspond to a difference up to 65%–

75% depending on particle size. Again, it should be remarked that similar to activated fractions, the residual GF distributions ($c_{res}$) can appear locally negative if the interstitial concentrations are higher than the respective total concentrations. Before converting these distributions into GF-PDFs shown in Fig. 1, the negative values were set to zeros which eventually strengthened the positive peaks appearing in normalized distributions. As a result, for example the less hygroscopic particle mode appearing in the estimated residual aerosol can be partially attributed to methodological uncertainties.

To our knowledge, this is one of the very few studies characterizing the hygroscopic properties of different in-cloud aerosol populations. Previously, Svenningsson et al. (1994) studied the aerosol hygroscopicity and its relationship to cloud droplet activation at Kleiner Feldberg in Germany. Interstitial aerosol hygroscopicity was measured during cloud events, and by assuming that the air mass was the same, it was compared to the total aerosol sampled during the following clear sky conditions. The less hygroscopic particle fraction was substantially higher in the interstitial population, indicating that the more hygroscopic particles were scavenged into the cloud droplets more efficiently than the less hygroscopic ones. This observation was confirmed in a case study, where a counter flow virtual impactor (CVI) was used to separate the cloud droplets from the total aerosol, so that the hygroscopicity of cloud droplet residuals could be measured independently. Likewise, Rose et al. (2013) observed a decreasing fraction of available CCN during a precipitative cloud event, suggesting that the more hygroscopic particles were mostly activated into cloud droplets and removed from the air through precipitation. Overall, theour results by Svenningsson et al. (1994)from Puijo are in a good agreement with ourthe observations from Puijo.
[revised manuscript text omitted]